# Phosphoproteomics of aged insulin-resistant bone identifies P70S6K phosphorylation of AFF4 as a gene-specific transcriptional regulator

Mriga Dutt [1,2], Luoping Liao[1], Hani Jieun Kim [3,4,5,6], Ronnie Blazev [1,2], Audrey Chan[1,2], Hitesh Kore[1,2], Ayenachew Bezawork-Geleta[1], Li Dong[1], Isela Sarahi Rivera[7,8], Natalie K. Y. Wee[9], Jeffrey Molendijk[1], Julian P. H. Wong [1,2], Vanessa R. Haynes[1], Veronica Uribe[1], Gordon S. Lynch [1,2], Kelly A. Smith [1], Magdalene K. Montgomery [1], Matthew J. Watt [1], Pengyi Yang [3,4], Garron T. Dodd [1], Stephin J. Vervoort [7,8], Natalie A. Sims [9] & Benjamin L. Parker [1,2] ✉

Insulin action on the skeleton is essential for bone development and whole-body energy metabolism, however a global view of signaling in this tissue is lacking. Furthermore, whether there are signaling differences that drive the gene-specific activation under insulin-resistant (IR) or ageing conditions is unknown. Here, we perform a phosphoproteomic analysis of insulin signaling in the bones of young, lean, insulin-sensitive versus old, obese, IR mice revealing a rewiring of phosphorylation. We target dysregulated phospho-proteins in a zebrafish functional genomic screen of bone development and mineralization revealing candidates important for skeletal formation. One of these is ALF Transcription Elongation Factor 4 (AFF4), the core scaffold of the Super Elongation Complex and we show that phosphorylation of S831 on AFF4 is an insulin-dependent substrate of P70S6K and attenuated in aged, IR bone. Phosphorylation of S831 is defective in IR osteoblasts and associated with reduced transcriptional elongation at discrete locations in the genome. Mechanistically, we show phosphorylation of S831 increases recruitment of chromatin remodelers, ENL/AF9 to crotonylated histone via the YEATS domain, and promotes gene-specific activation. Our analysis identifies regulators of insulin action on the skeleton, further uncovering a mechanism of IR via locus-specific changes in transcriptional elongation and gene activation.

Insulin signaling plays an important role in bone and has effects on both skeletal development, remodeling and energy metabolism. Early calvarial bone organ culture studies showed that treatment with insulin increased collagen production and promoted glucose uptake[1–3]. Mice with conditional Insulin Receptor (InsR) deletion in osteoblasts and osteocytes have low trabecular bone mass, accompanied by lower bone formation rate and low osteoblast numbers compared to controls[4]. These mice are also obese and have lower β-cell

mass, insulin content and attenuated glucose-stimulated insulin secretion resulting in low blood insulin levels, high fasting blood glucose levels, and glucose intolerance[4,5]. The effects of insulin signaling within the skeleton and the subsequent systemic responses are thought to be mediated by a suite of transcriptional networks and circulating factors secreted specifically by bone (for a review see ref. 6).

A key question is: how does insulin regulate the expression of specific genes? This could occur at each of the main steps in the transcriptional cycle: (i) Initiation: transcription factors are recruited and enable RNA-Polymerase-II (RNAP2) binding to the DNA promoter region, (ii) Elongation: the bound and paused RNAP2 is released to travel along and transcribe DNA, and (iii) Termination: the nascent mRNA is cleaved and polyadenylated[7]. Insulin-regulated gene expression has primarily been studied at the level of transcriptional initiation, including modulation of transcription factor activity by PI3K/Akt/mTORC1 and MAPK pathways, but the most important pathways for insulin action in bone remain elusive. For example, there has been some evidence that the effects of insulin on bone are dependent on FOXO1, since deletion of a single copy of *Foxo1* in osteoblast-specific InsR knockout mice partially rescued glucose intolerance, and normalized the bone resorption marker CTx, but the bone phenotype of these mice was not reported[5]. Activator Protein-1 (AP1) complexes comprising homo- or hetero-dimers of JUN, FOS or ATF family transcription factors are activated following insulin stimulation by MAPK phosphorylation leading to expression of Immediate Early Genes (IEGs) such as Early Growth Response Factor 1 (EGR1) which is important for cell growth and proliferation[8–10]. Mice lacking JNK1/2 specifically in the osteoblast lineage have reduced AP-1 activation, defects in osteoblast differentiation and mineralization capacity, and display severe osteopenia[11]. Furthermore, knockdown of *Egr1* in osteoblasts reduces BMP-induced mineralization[12], and EGR1 cooperates with RUNX2 to promote the expression of osteoblastic genes such as *Osterix* and *Osteocalcin*[13]. The relative contribution and capacity of these pathways to functional insulin signaling in bone is unknown.

Despite most studies focusing on insulin-dependent transcriptional initiation, the release of paused RNAP2 into active transcriptional elongation has now been identified as the dominant rate limiting step of gene expression (for a review see ref. 14). This is because for the majority of protein-coding genes, RNAP2 is paused proximal to the promoter where it remains bound to DNA, poised and awaiting further signals to resume elongation and complete RNA transcription[15]. This mechanism enables rapid and synchronized release of RNAP2 for robust gene expression in response to external signals or stress and can be quickly turned off via negative feedback[16]. Release of paused RNAP2 into full RNA synthesis is triggered by binding of the Super Elongation Complex (SEC) which has ALF Transcription Elongation Factor 4 (AFF4) as its core scaffold and recruits the kinase activity of Positive Transcription Elongation Factor-b (P-TEFb) which phosphorylates the C-terminal domain (CTD) of RNAP2, Negative Elongation Factor (NELF) and DRB Sensitivity-Inducing Factor (DSIF) (for a review see ref. 17). There is growing evidence to suggest that extracellular stimuli or growth factors can promote gene expression via acute phosphorylation of transcriptional elongation factors. For example, EGF stimulation activates ERK-dependent phosphorylation of NELF-A and enhances elongation of specific IEGs[18]. It is not known whether insulin regulates the release of paused RNAP2 into active transcriptional elongation to promote gene expression via similar or alternative mechanisms.

Since the discovery of insulin by Macleod, Banting, Best and Collip in the 1920's, the vast majority of studies on global insulin signaling have focused on tissues such as skeletal muscle[19–21], liver[22–24], and adipose[25–27]. These studies have provided panoramic views of the insulin signaling landscape but, despite the importance of insulin

signaling in bone on skeletal development and whole-body energy metabolism, a similar analysis in this tissue has not been performed. Hence, the goal of the current study was to map the in vivo phosphorylation-based insulin signaling network in bone and identify; (i) phosphorylation sites that may be dysregulated in an age-dependent insulin-resistant model, (ii) phosphoproteins that may be contributing to bone development and mineralization, and (iii) functional phosphorylation sites that contribute to insulin-dependent gene activation and proteome remodeling. In this study, we performed the first global phosphoproteomic and proteomic analysis of insulin signaling in bones from young, lean and insulin-sensitive versus old, obese and insulin-resistant (IR) mice revealing changes in kinase activity and hundreds of dysregulated phosphosites. We prioritized candidates based on several integration strategies, and this target list was used to develop a functional genomic screen of bone growth, development and mineralization in zebrafish. This identified regulators of skeletal development including AFF4 which we show is a P70S6K substrate that is defective in insulin-resistant osteoblasts and associated with a reduction in transcriptional elongation of specific genes. Mechanistically, we show that phosphorylation of AFF4 recruits ENL/AF9 to the SEC and fine-tunes insulin stimulated gene-specific activation, independently of the P-TEFb. This work identifies several regulators of insulin action in the skeleton including a mechanism of insulin resistance via aberrant P70S6K-mediated phosphorylation of AFF4, attenuated transcriptional elongation and defective gene activation.

## Results

### Phosphoproteomic analysis of insulin-dependent and age-related changes in bone

We compared the in vivo insulin signaling response in bone tissue of 10- versus 73-week-old C57BL/6 J male mice using multiplexed stable isotope-based phosphoproteomics and proteomics (Fig. 1A). The 73-week-old mice had lower lean mass consistent with age-associated sarcopenia and a > 3-fold increase in whole-body adiposity (Supplementary Fig. 1). These aged mice also had elevated fasting blood glucose and insulin translating to an increase in the Homeostatic Model Assessment for Insulin Resistance (HOMA-IR) compared to 10-week-old mice. Mice were fasted for 12 h and saline or insulin (2.5 mU/kg) was interperitoneally injected, and mice were sacrificed after 20 min. Tibiae were rapidly dissected and flushed with ice-cold PBS within 30 s to remove marrow and quench phosphorylation. Consistent with their aged phenotype, tibiae from 73-week-old mice had lower cortical thickness, trabecular bone volume and trabecular number, and with a greater marrow area, trabecular separation and thickness than tibiae from 10-week-old mice (Fig. 1B). In the insulin-stimulated bones, western-blot analysis confirmed increased phosphorylation of S473 on Akt kinase and S235/6 on Ribosomal Protein S6 (RPS6), two known signaling events of the insulin signaling pathway (Fig. 1C). While insulin-dependent phosphorylation of S473 on Akt was similar in both age groups, the phosphorylation of RPS6 was attenuated in the bones of 73-week-old mice. This age-related reduction in the response of bone to insulin was confirmed by western-blot analysis using a pan Akt/P70S6K substrate antibody which binds phosphorylation in the kinase motif RXRXXpS/T (Fig. 1C).

Phosphoproteomic analysis of tibiae identified 19,801 unique phosphopeptide sequences and 16,502 class I phosphosites ( > 0.75 phosphorylation localization probability to a Ser, Thr or Tyr) on 4,803 phosphoproteins (Supplementary Data 1). We first focused our quantitative analysis on the insulin response in 10-week-old bone and identified 1670 insulin-regulated phosphopeptides (1418 phosphosites with >0.75 localization probability) ($q < 0.1$; limma moderated t-test with Benjamini Hochberg FDR) (Fig. 1D). Phosphorylation of several substrates of Akt, mTOR and P70S6K were increased such as S210 on AKT1S1, T45 on EIF4EBP1, and S422 on EIF4B, respectively, confirming

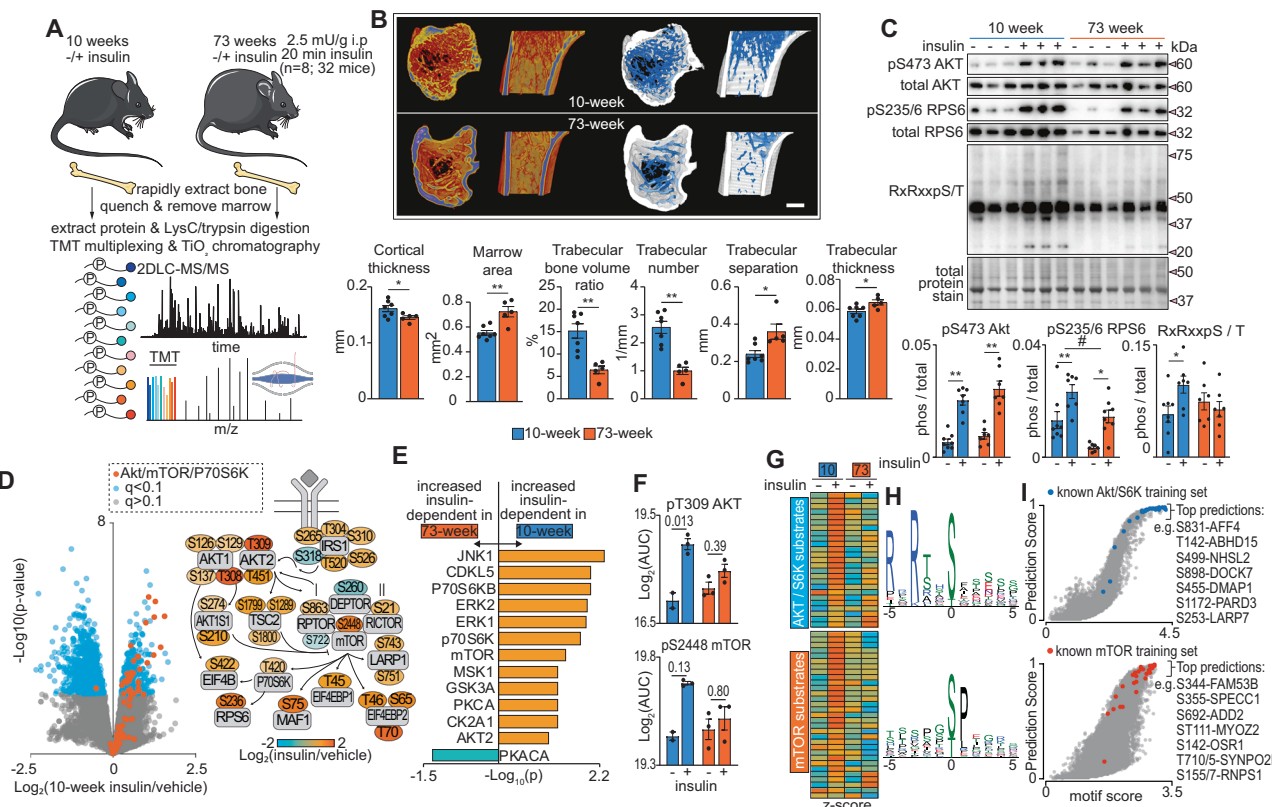

**Fig. 1 | Insulin- and age-associated phosphoproteome of mouse bone.**
**A** Experimental design of the phosphoproteomic analysis of insulin signaling in 10- and 73-week-old bone. Illustration modified from Servier Medical Art licensed under Creative Commons Attribution 4.0 International (CC BY 4.0) (https://creativecommons.org/licenses/by/4.0/). **B** Microcomputed tomography representative images of tibia bone and quantification summary ($n = 5$-7 biological replicates, $*p < 0.05$, $**p < 0.01$, unpaired t-test scale bar = 500 μm). **C** Western-blot analysis of bone lysates for known insulin-regulated phosphorylation events (n = 7 biological replicates, $*p < 0.05$, $**p < 0.01$, unpaired $t$-test; $\#p < 0.05$, two-way ANOVA). **D** Volcano plot showing the phosphopeptide $Log_2$ fold-change (insulin/vehicle) of 10-week-old bone plotted against the $-Log_{10}$ $p$-value highlighting regulated phosphopeptides (blue) and known Akt, mTOR or P70S6K substrates (red). Shown on the right is a representation of the insulin signaling pathway with

phosphosite quantification. **E** Kinase enrichment analysis showing kinases with increased insulin activation in 10-week (orange) or 73-week-old bone (green) ($p < 0.1$, Fisher's Exact test). **F** Quantification of regulatory phosphosites on Akt and mTOR ($n = 3$ biological replicates) with indicated unpaired t-test q-values adjusted for multiple hypothesis testing using Benjamin Hochberg FDR. **G** Heatmap showing the quantification of insulin-regulated phosphorylation in 10- and 73-week old bone that are known Akt/P70S6K and mTOR substrates, and used as the training set for machine learning predictions. **H** Motif enrichment analysis of amino acids surrounding the phosphorylation sites of known AKT/P70S6K and mTOR. **I** Machine learning prediction of candidate kinase substrates highlighting known Akt/P70S6K (blue) or mTOR (red) training set. Data are presented as +/− standard error of the mean. Source data are provided as a Source Data file.

the utility of our bone phosphoproteomic data. To investigate signaling differences between 10- and 73-week-old bone, we first compared the vehicle- and insulin-treated groups. Where possible, this was normalized to the abundance of the total protein levels, since we saw widespread changes in the proteome between the age groups (see below). Such analysis has previously been performed to compare phosphoproteome versus proteome responses to acute insulin stimulation in the setting of chronic treatments that induce insulin resistance[27]. Of the 4,803 phosphoproteins, we obtained proteomic data for 3,195 proteins and normalized the abundance of 15,642 phosphopeptides to their associated protein abundance. In the vehicle-treated groups, there were 65 differentially regulated phosphopeptides between 10-week and 73-week-old mice but when normalized to total protein levels, none of these remained significant ($q < 0.1$; limma moderated $t$-test with Benjamini Hochberg FDR). This contrasted with the insulin-stimulated tibiae, where 5471 regulated phosphopeptides differed between 10-week and 73-week-old mice with 650 of these being differentially regulated when normalized to total protein levels. We next compared the magnitudes of the insulin response i.e., difference of vehicle versus insulin stimulation between the age groups. Remarkably, of the 1418 insulin-regulated phosphosites in young bones, 1004 showed differences in the magnitude of

their response in the 73-week-old mice with 37% of these showing a lower insulin response and 63% showing greater responses ($q < 0.05$; limma moderated t-test with Benjamini Hochberg FDR)(Supplementary Data 1). This increased magnitude of phosphorylation with age-related IR in bone is consistent with the growing evidence that hyperactivation or 'emergent' phosphorylation is more pronounced in IR[27].

We next investigated kinase activity by performing an enrichment analysis on known kinase-substrate relationships (KSRs) retrieved from the PhosphositePlus database[28]. Of the 16,502 phosphosites quantified, only 459 (2.78%) were annotated with a known upstream kinase. Despite this low annotation, substrates of known kinases displayed differences in insulin-induced phosphorylation between the age groups ($p < 0.1$, Fisher's Exact test) (Fig. 1E). Several kinases displayed greater insulin-dependent activity in 10-week vs 73-week-old bone including Akt, P70S6K, mTOR and ERK1/2. A relative reduction in the insulin-stimulated activity of Akt and mTOR kinases was inferred by lower insulin-stimulated phosphorylation of the activating phosphosites T308/9 on AKT1/2 and S2448 on mTOR in the 73-week bones compared to the 10-week bones (Fig. 1F). PKA was the only kinase enriched in the 73-week-old bones and suggests insulin's ability to suppress PKA signaling was attenuated with age. These data support

altered kinase activity following acute insulin stimulation in bone tissue of aged and IR mice.

## Predicting KSRs of Akt/P70S6K and mTOR

Given the low number of phosphosites annotated with known kinases, we used a previously described machine learning approach[29,30] to predict additional KSR substrates for Akt/P70S6K and mTOR given their altered activity in the 73-week-old bones. A training set of known substrates were compiled for each kinase and used to train base classifiers of support vector machines using both the phosphorylation expression profiles between the four groups (Fig. 1G), and the amino acid motif surrounding the phosphosite, a feature that confers kinase specificity (Fig. 1H) (Supplementary Data 1)[31]. This generates a motif and final prediction score ranging from 0 to 1 (Fig. 1I). As expected, many high-ranking phosphosites were known substrates of the kinases used to train the model (indicated as blue and orange dots in top right of plots in Fig. 1I). Other top-ranked phosphosites were previously predicted as Akt/P70S6K or mTOR substrates and are insulin-responsive in 3T3-L1 adipocytes[29]. For example, previously predicted Akt substrates included S499 on Nance-Horan Syndrome Like protein-2 (NHSL2), a protein involved in actin-cytoskeleton remodeling; T142 on Alpha/beta Hydrolase Domain-containing Protein 15 (ABHD15), a protein associated with insulin-mediated suppression of PKA signaling and lipolysis[32]; and several sites on Dedicator of Cytokinesis protein-7 (DOCK7), a guanine nucleotide exchange factor that has also been associated with lipid metabolism in a human genome wide association study (GWAS)[33]. We also identify uncharacterized insulin-responsive kinase predictions. For example, phosphorylation of S831 on AFF4 was identified as an insulin-dependent phosphosite located within an Akt/P70S6K kinase consensus motif with a kinase prediction score of 0.99. AFF4 is the core scaffold of the SEC that controls the release of paused RNAP2 into active transcriptional elongation. Variants in *AFF4* have been implicated in the rare genetic syndrome CHOPS where patients show shortened stature, facies, skeletal dysplasia and obesity[34,35]. Another interesting prediction was phosphorylation of S155/7 on RNPS1 which had a mTOR prediction of 0.85. RNPS1 is important for alternate splicing and nonsense-mediated mRNA decay[36] and has been identified as insulin-responsive in mouse liver[23] but it's functional importance in bone is unknown. Taken together, our phosphoproteomic analysis has identified divergent phosphorylation responses and kinase activities. Our results lay the foundations to validate upstream kinase catalysts and prioritize functional studies of the phosphoproteome to mechanistically understand how insulin action on the skeleton contributes to bone development, ageing physiology and insulin resistance.

## Defining the mouse bone-enriched proteome and its age-associated regulation

We identified 7529 protein groups (6046 with two or more peptides) within the bone proteome spanning 7374 gene entries. This is >3-fold the coverage of a recent deep bone proteomic study, likely made possible by our use of peptide-level UHPLC fractionation prior to LC-MS/MS[37] (Supplementary Data 2). Since neither the Human Protein Atlas[38] nor mouse proteomic tissue atlases[39,40] include bone tissue, we used this to generate a draft list of the enriched mouse bone proteome. To achieve this, we used intensity-based absolute quantification (iBAQ)[41] and normalized our current proteomic data and previously published proteomic data from 28 mouse tissues[40] to PARK7, a protein reported to have the lowest variability in abundance amongst a variety of cell types and tissues[42]. We then calculated fold-changes for each protein in bone versus the 28 tissues. We used criteria established in the Human Protein Atlas based on >4-fold higher levels in bone compared to all other tissues to identify 292 bone enriched proteins (Fig. 2A). Pathway analysis of this enriched bone proteome identified known proteins with functional roles

in skeletal morphogenesis, osteoblast differentiation and several collagens and collagen-associated proteins such as Prolyl 4-hydroxylase alpha-2 (P4HA2), which catalyzes the hydroxylation of collagen. The enriched bone proteome also identified proteins with unclear mechanisms in bone. For example, Quiescin Q6 Sulfhydryl Oxidase-1 (QSOX1), a protein involved in extracellular disulfide bond formation was 163-fold enriched in bone compared to the other tissues. Interestingly, recent data from the International Mouse Phenotype Consortium[43] suggests whole-body homozygous *QSox1* KO mice display reduced bone mineral content in both male and female mice.

We next focused our analysis on a quantitative comparison of the bone proteome between the vehicle- and insulin-treated 10- or 73-week-old mice. Only a single protein (Smoothelin-like Protein-2; SMTNL2) displayed differential abundance between vehicle and insulin groups. This was unsurprising given the short 20 min stimulation, which is unlikely to be long enough to observe in vivo changes in protein abundance across the entire tissue. We therefore combined the vehicle- and insulin-stimulated mice in their respective age groups to increase statistical power to identify differences associated with age. This identified 2784 protein groups with differential abundance in the bones of 10- versus 73-week-old mice ($q < 0.01$; limma moderated t-test with Benjamini Hochberg FDR) (Supplementary Data 2). Gene set enrichment analysis (GSEA) revealed lower ribosome and spliceosome content in aged bones consistent with previous proteomic analysis of several aged tissues[44] (Fig. 2B-C). While we observed a general decrease in protein degradation systems including the proteasome and lysosome, we found elevated levels of lipid/amino acid degradation systems such as the peroxisome. The bones of aged mice also had greater levels of several enzymes regulating glucose and fatty acid metabolism, again consistent with other aged tissues, including Carnitine O-palmitoyltransferase 1 (CPT1B) which regulates mitochondrial long-chain fatty acid uptake, was significantly up-regulated in bone and previously identified as a cross-tissue enriched age- associated protein[44]. Retinol metabolism was also significantly elevated in aged bone with the most elevated protein in this pathway being Aldehyde dehydrogenase 1A1 (ALDH1A1), which catalyzes retinaldehyde oxidation into retinoic acid. This is interesting because population studies have suggested an association between retinol consumption and fracture risk[45-48]. This is also interesting because ALDH1A1 displays tissue-specific age associations with significant up-regulation in kidney, skeletal muscle, and adipose but remains unchanged in liver and brain[44].

To understand the potential transcriptional mechanisms controlling the age-associated bone proteome, we next mapped proteins to known upstream transcriptional regulators in the ChEA3db[49] and performed GSEA (Fig. 2D). Here, we focused on transcriptional regulators that potentially display defective activity and may contribute to decreased protein abundance in aged bone. The most significantly enriched transcriptional regulator was RE1-Silencing Transcription Factor (REST), which represses neuronal gene transcription in non-neuronal cells and is dysregulated in several cancers. The role of REST in bone is relatively unexplored but it is required for osteoblast differentiation in vitro[50]. Other interesting transcriptional regulators include ETS-related transcription factor (ELF1) and Transcriptional Repressor Protein (YY1) that both had decreased protein abundance in 73-week compared with 10-week-old bone. Target genes of FOS and JUN, both downstream transcriptional regulators of insulin signaling, were also significantly decreased suggesting defective insulin-induced activation in aged bone (Fig. 2E-F). In summary, our analysis has defined the C57BL/6 J male mouse bone proteome identifying enriched proteins in the skeleton, and defined age-associated pathways including defective upstream transcriptional regulators and downstream metabolic pathways and secreted factors.

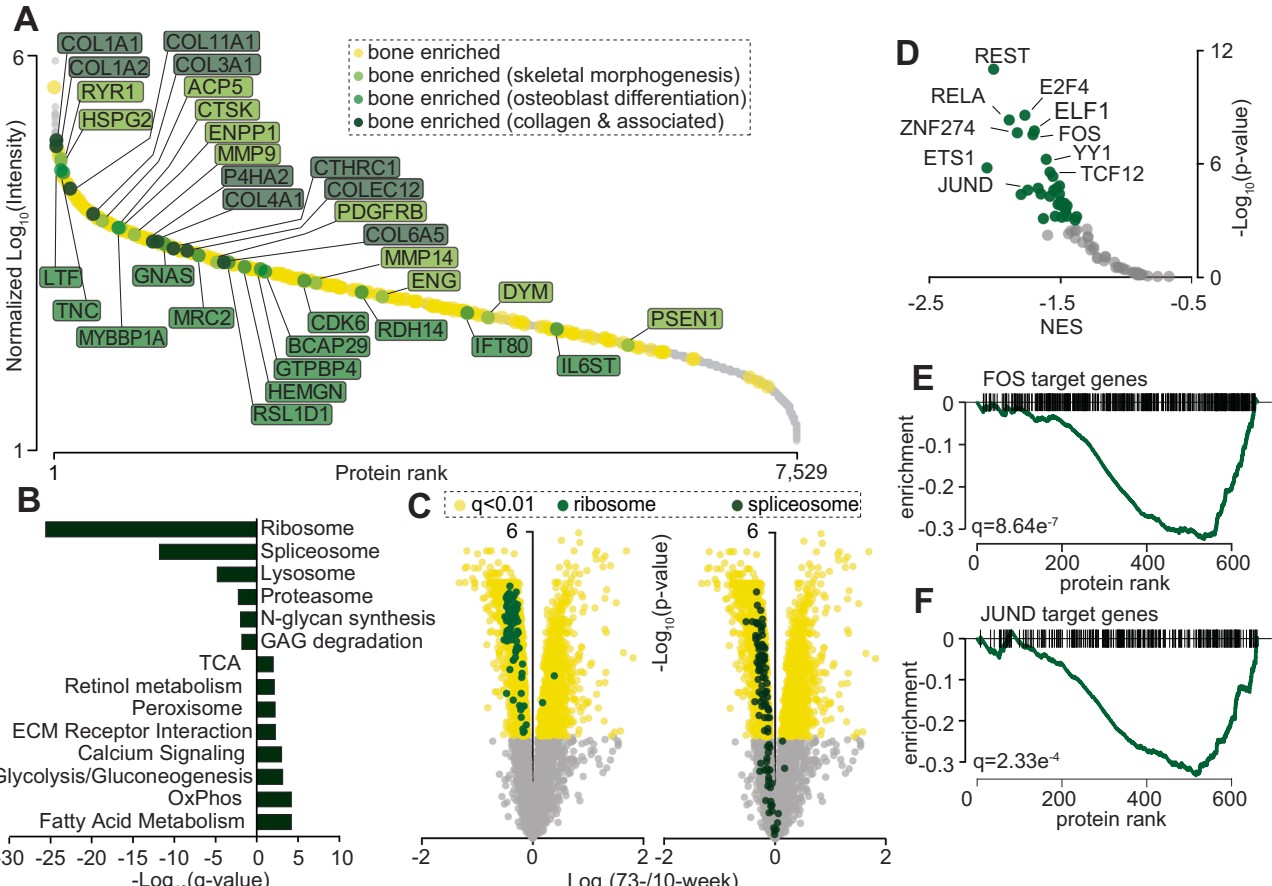

**Fig. 2 | The bone-enriched proteome and regulation during ageing. A** Ranked abundance of the mouse bone proteome highlight tissue enrichment. **B** Gene set enrichment analysis (GSEA) of KEGG pathways significantly regulated in 10-week vs 73-week-old mouse bone. Left = enriched in 10-week-old bone; right = enriched in 73-week-old bone. **C** Volcano plot showing the protein Log2 fold-change (73-week/ 10-week) plotted against the -Log10 *p*-value highlighting ribosome and spliceosome KEGG annotations proteins. **D** GSEA of the proteomics data using the transcription factor:target geneset annotations in the ChEA3 database. Data presents the normalized enrichment score (NES) focusing on proteins down-regulated in the 73-week old bone to estimate potential decreased transcription factor activity. Individual GSEA plots of the target genes for FOS (**E**) and JUND (**F**) showing the target genes are significantly down-regulated where proteins are ranked from up-regulated to down-regulated. Source data are provided as a Source Data file.

## Functional genomic analysis of insulin-dependent phospho-proteins in zebrafish identifies AFF4 as a regulator of bone development

Our phosphoproteomic and proteomic analyses of insulin signaling in aged- and IR bone identified hundreds of dysregulated phosphoproteins which may contribute to skeletal development/remodeling, ageing physiology and/or whole-body energy metabolism. To prioritize candidates for assessing their causal role in bone biology, we devised a three-step integration approach. First, we identified proteins containing insulin-regulated phosphorylation in 10-week-old bones and mapped their encoding genes to a recent large-scale human genome-wide association study (GWAS) of bone mineral density (BMD) which identified 133 candidates[51] (Fig. 3A) (Supplementary Data 1). Next, we identified highly conserved phosphosites between mouse and zebrafish by performing a phosphoproteomic analysis of zebrafish caudal fin clippings. We identified 20,460 unique phospho-peptides (16,699 class I phosphosites) of which 1977 phosphosites on 1363 proteins were highly conserved in the mouse bone phospho-proteome (Supplementary Data 1 and 3). Finally, we integrated these data with the machine learning kinase-substrate predictions of insulin-dependent phosphosites of Akt/P70S6K and mTOR (Prediction Score>0.75) (Supplementary Data 1). We refined candidate genes that were overlapping in at least two out of the three selection criteria, and further filtered for genes containing differences in the their magnitude

of insulin-dependent phosphorylation in 10- verses 73-week old mice with signaling and proteome remodeling potential e.g., transcription factors, translation/splicing regulators or kinases resulting in a list of 22 candidate genes. These were targeted in a loss-of-function screen in zebrafish for assessment of body growth, skeletal development and mineralization. One-cell staged embryos were injected with a Cas9/ 4gRNA RNP targeting the candidate genes or a paired negative scramble (scr) control. The screen also included three known regulators of bone development and bone mineralization: *akt2*, *insr* and *entpd5*. The generated F0 crispants underwent Alizarin red mineralization staining, imaging, and PCR-based genotyping at 21 days post fertilization (dpf) to identify bone-modulating phosphoproteins. All positive controls displayed drastic reductions in growth and/or mineralized area consistent with the literature, confirming the effectiveness of our genetic knock-down approach[52–54] (Fig. 3B and Supplementary Data 4). Knockdown of 4 genes (*cx43*, *fcho2*, *gps1* and *med12*) were embryonic lethal (Fig. 3B).

Twenty seven percent (6/22) of the candidate genes emerged as positive hits of the functional genomic screen with crispants displaying bone phenotypes that were significantly different to the paired scr (Fig. 3B and Supplementary Data 4). Knockdown of two genes, *autophagy associated transmembrane protein (ei24)* and *stard3 n-terminal like (stard3nl)*, resulted in drastic global growth defects. *ei24* crispants displayed a severely compromised growth pattern (Fig. 3B–C

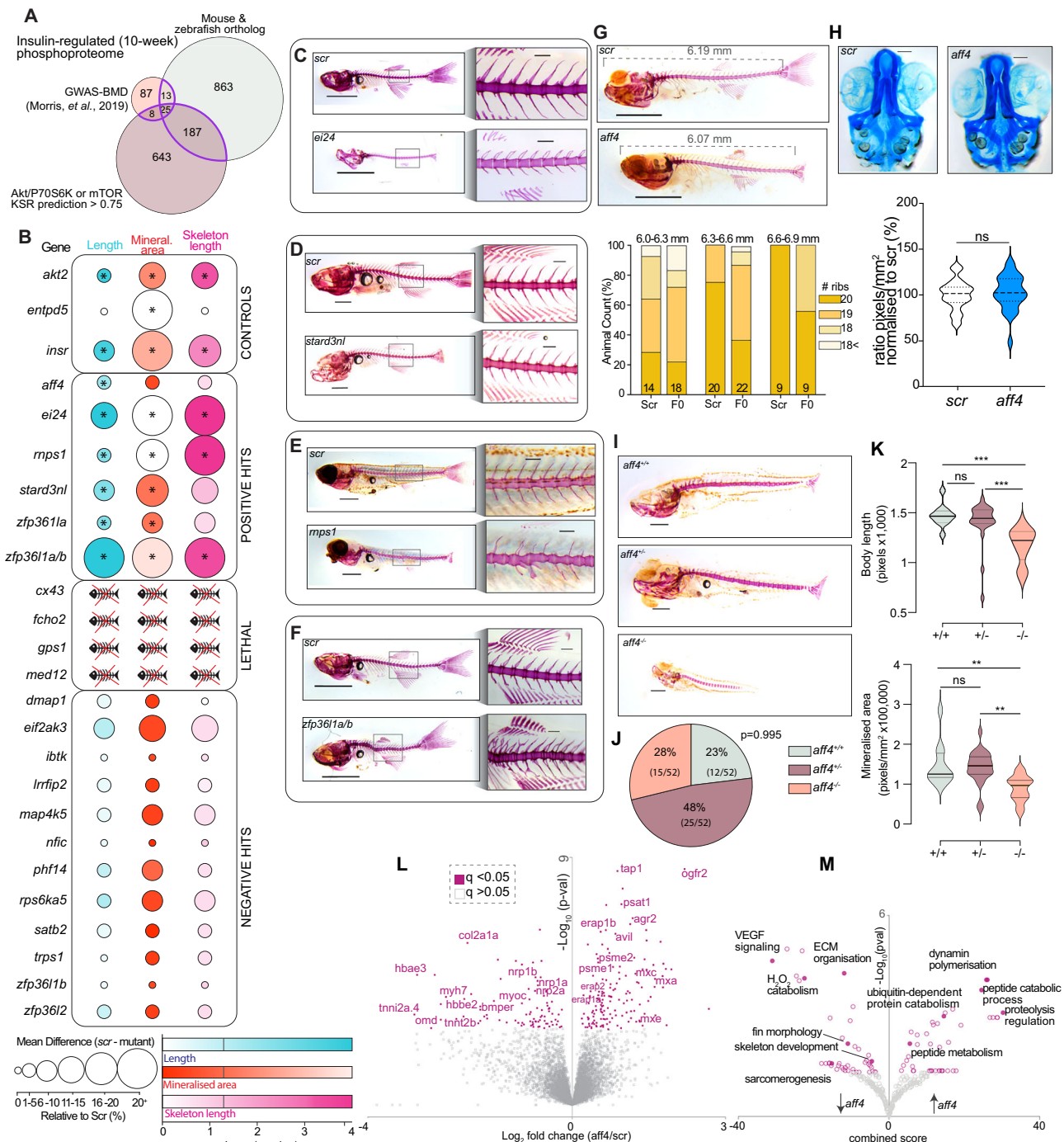

**Fig. 3 | Insulin-regulated phosphoproteins targeted in a functional genomic analysis of body growth, bone mineralization and skeleton development in zebrafish. A** Schematic representation of the three-step candidate prioritization approach. **B** Quantification of body length, alizarin red mineralization staining, and skeleton length. Significant datasets are denoted by an asterisk (*). Representative images of paired scramble gRNA mutants (scr), and crispants of (**C**) *ei24* (scale bar = 1 mm), (**D**) *stard3nl* (scale bar = 1 mm), (**E**) *rnps1* (scale bar = 1 mm), and (**F**) *zfp36l1a/b* (scale bar = 1 mm). Zoom insert scale bar = 0.2 mm. **G** Representative image of paired scr and *aff4* crispants (scale bar = 1 mm), and size binning rib count analysis. **H** Quantification of head cartilage using alcian blue. Representative images of

paired scr and *aff4* crispants (ns= $p > 0.05$, unpaired two-sidedt-test; scale bar = 0.1 mm). **I** Representative images of *aff4* germline heterozygous and homozygous mutants with paired wild-type sibling (scale bar = 0.5 mm). **J** Ratios of *aff4* germline mutant offspring ($p = 0.995, X^2$ test) and (**K**) quantification of body length and bone mineralized area (*$p < 0.05$, **$p < 0.01$, ***$p < 0.001$, One-way ANOVA). **L** Volcano plot showing significant, regulated proteins ($q < 0.05$, Benjamini–Hochberg correction) and (**M**) the associated pathway enrichment analysis of GO-Biological Processes ($p < 0.05$, Fisher's exact test) in the *aff4* knockdown caudal fin proteome. Source data are provided as a Source Data file.

and Supplementary Data 4) including a 32% and 36% reduction in mineralized area and skeleton length, respectively. *ei24* is primarily involved in cellular autophagy to remove protein aggregates and help maintain cellular protein cargo homeostasis[55]. A role for *ei24* has

emerged in osteogenesis and frameshift mutations in this gene have been linked to the development of osteosarcoma[56]. Similarly, *stard3nl* crispants displayed 9%, 18% and 15% reduction in body growth, mineralized area and skeleton length, respectively (Figs. 3B, 3D and

Supplementary Data 4). This is intriguing as *stard3nl* is a known GWAS loci for BMD[57] and has been shown to be a negative regulator of osteogenesis by inhibiting the WNT/β-catenin pathway[58]. Two RNA-binding proteins emerged as positive hits from the screen. First, *rnps1* crispants exhibited a striking phenotype of lacking ribs along the axial skeleton with a 20% and 24% reduction in mineralized area and skeleton length, respectively (Figs. 3B, 3E and Supplementary Data 4). These findings provide functional evidence of a role of *rnps1* in bone biology. The second gene was the *zinc finger protein 36, c3h type like 1*(*zfp36l1*), which has two paralogues in zebrafish, *zfp36l1a* and *zfp36l1b*. *zfp36l1* is suggested to be involved in osteoblast differentiation processes mediated by the parathyroid hormone receptor that undergoes senescence-triggered defects[57]. While individual knockdown of *zfp36l1a*, but not *zfp36l1b*, resulted in a modest reduction of growth and mineralized area (Supplementary Data 4), knockdown of both genes resulted in crispants with 45% reduced mineralized area and striking skeletal dysplasia including scoliosis (Figs. 3B, 3F and Supplementary Data 4).

The final positive hit from this screen was the transcription factor *aff4* where crispants exhibited significantly reduced growth (Fig. 3G). Given zebrafish development is proportional to its standard body length[59,60], we sorted them into three size bins and checked for developmental differences by performing a total rib count (Supplementary Data 5). *aff4* crispants within each size bin had fewer ribs compared with scr crispants of the same body length (Fig. 3G), suggesting *aff4* knockdown delays zebrafish bone development. These data combined with the previously mentioned clinical features of patients with CHOPS syndrome displaying impairments in skeletal development and obesity caused by missense variants in *AFF4* promoted us to further expand our analysis[34,35]. To assess if developmental defects of the skeletal system were specific to bone, we next performed alcian blue staining revealing no difference in cartilage development (Fig. 3H). The F0 *aff4* crispants were next backcrossed onto the wild-type AB strain to generate F1 heterozygotes that were then in-crossed to generate F2 germline mutants which followed expected Mendelian ratio at 21 dpf (Fig. 3I–J). Homozygous *aff4* knockouts displayed severely reduced whole-body growth and bone mineralized area at 21 dpf (Fig. 3K). Proteomic analyses of the caudal fins revealed significant regulation of 237 proteins in the *aff4* knockdown fish when compared to the scr control ($q < 0.05$; Benjamini-Hochberg correction) (Fig. 3L and Supplementary Data 6). Pathway enrichment analysis using FishEnrichr revealed an up-regulation of proteins involved in ubiquitin-dependent proteolysis and associated peptide catabolic processes (Fig. 3M). These include the proteosome activator complex subunits 1/2 (PSME1/2) and the interferon-induced GTP-binding protein class (MXA, MXC, MXE). Knockdown of *aff4* was associated with the down-regulation of proteins involved in VEGF signaling, ECM organization and the development of the skeleton and fin morphology such as Osteomodulin (OMD) and Myocilin (MYOC), both positive regulators of osteoblast differentiation, along with type II collagens (COL2A1A and COL2A1B) that are required for skeletal growth. Additionally, proteins involved in VEGF signaling important for angiogenesis (Neuropilins, NRP1A, NRP1B, NRP2A; BMP-binding endothelial regulator protein precursor, BMPER), and systemic muscle development (myosin heavy chain 7,MYH7; cardiac troponin T type 2B,TNN2B) were observed to be significantly down-regulated, suggesting that *aff4* knockdown fish have a compromised musculoskeletal development when compared to the scr. Altogether, our targeted functional genomic analysis of insulin-regulated phosphoproteins identified several regulators of development and mineralized area, including AFF4, which we validated through genetic and proteomic analyses as required for zebrafish skeletal formation. Given AFF4's role as a core scaffold of the SEC and importance in regulating transcriptional elongation, we next characterized the upstream kinase(s) and whether phosphorylation regulates insulin-dependent gene activation and aberrant transcriptional responses.

## S831 on AFF4 is a P70S6K substrate

Phosphorylation of AFF4 was identified by LC-MS/MS in mouse bone (S831) and zebrafish tails (S771) with insulin-dependence located within an Akt/P70S6K kinase consensus motif and a kinase prediction score of 0.99 (range 0–1) (Fig. 4A). The phosphorylated amino acid and surrounding kinase motif is highly conserved across animalia kingdom, an important feature of previously characterized functional phosphorylation sites[61]. Normalized phosphorylation intensity was significantly attenuated in the insulin stimulated bones from 73-week-old mice although the magnitude of the insulin response was similar to that of 10-week-old mice (Fig. 4B). To provide a deeper coverage of AFF4 phosphorylation, HEK293T cells expressing FLAG-tagged AFF4 were treated with or without insulin for 20 min followed by anti-FLAG immunoprecipitation and LC-MS/MS. A total of 47 phosphosites were quantified with only S831 identified as significantly regulated with insulin stimulation (Fig. 4A and Supplementary Data 7). To investigate Akt versus P70S6K phosphorylation of S831, HEK293T cells were treated with or without insulin in the presence of an Akt or P70S6K inhibitor. Insulin stimulation in the presence of the Akt inhibitor abolished phosphorylation of S473 on Akt and the direct Akt substrate T246 on PRAS40 (AKT11S1), as well as downstream S235/6 on RPS6, a direct substate of P70S6K (Fig. 4C–D). In contrast, insulin stimulation in the presence of the S6K inhibitor only abolished downstream phosphorylation of S235/6 on RPS6 while there was a significant increase in phosphorylation of S473 on Akt consistent with negative feedback on IRS1[62] (Fig. 4C–D). LC-MS/MS of immunoprecipitated AFF4 revealed inhibition of S831 phosphorylation by both Akti and S6Ki suggesting P70S6K-depedendent phosphorylation (Fig. 4D) (Supplementary Data 7). We also observed direct P70S6K phosphorylation of S831 on AFF4 via an in vitro kinase reaction with purified enzyme/substrate and analysis by LC-MS/MS (Fig. 4E).

Given the possibility that AFF4 phosphorylation is mediated by P70S6K phosphorylation, we then knocked-down P70S6K in zebrafish (double knockdown of *rps6kb1/2*) using embryo Cas9/gRNA RNP injections. This resulted in significant reductions in whole-body growth and skeletal development, as well as the abundance of P70S6K protein and the expected phosphorylation of S235/6 on RPS6 in zebrafish tail clippings (Fig. 4G–H). Phosphoproteomics of zebrafish tail clippings revealed a significant decrease in phosphorylation of S771 on AFF4 (S831 and S836 in mouse and human, respectively) along with S236/40 on RPS6 ($q < 0.05$, *t*-test with permutation-based FDR) (Fig. 4I–J and Supplementary Data 8). Through pharmacological, biochemical and genetic approaches, we demonstrate that S831 on AFF4 is a direct P70S6K substrate, both of which are required for normal skeletal development in zebrafish.

## Defective phosphorylation of S831 on AFF4 correlates with a reduction in the release of paused RNAP2 in IR osteoblasts

IR is associated with defective gene activation in response to acute insulin stimulation[63]. Under normal healthy states, acute insulin treatment increases mRNA levels of anabolic IEGs such as *Jun, Fus* and *Fos*, but the expression of these genes is markedly attenuated under IR states. Given that release of paused RNAP2 and mRNA elongation is a critical checkpoint for gene transcription, we hypothesized that attenuated insulin-dependent phosphorylation of S831 on AFF4 may underlie the transcriptional defects in IR conditions. To investigate this hypothesis, we established an osteoblast IR model by exposing differentiated Kusa 4B10 cells to chronic hyperinsulinemia. First, we characterized the model by quantifying the mineralization capacity, proteome changes and acute insulin signaling responses. Alizarin red staining revealed a significant decrease in the formation of mineralized nodules under IR conditions, a consistent phenotype observed

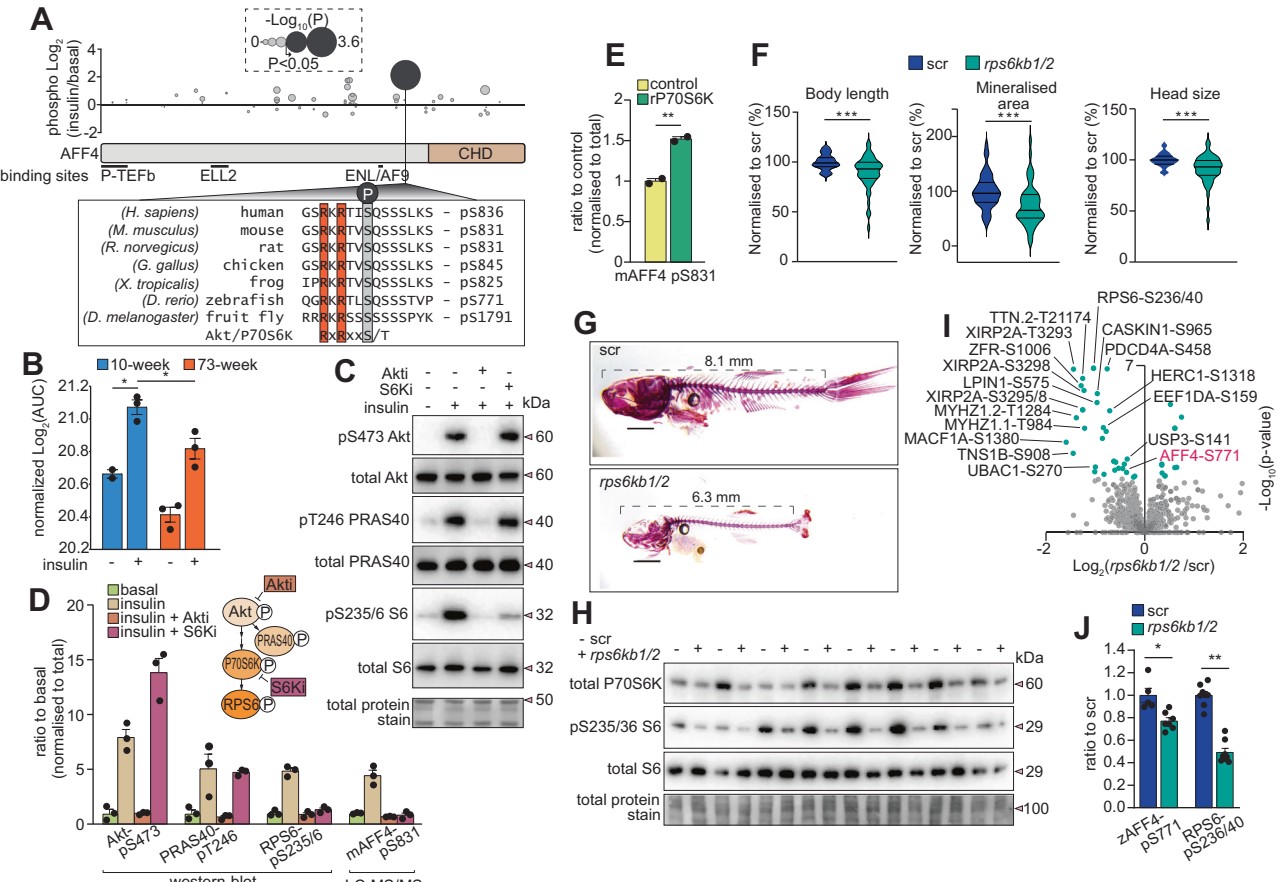

**Fig. 4 | AFF4 is a P70S6K substrate. A** Insulin-regulated phosphosite mapping of AFF4 and domain annotation with zoomed insert of multiple sequence alignment indicating the Akt/P70S6K motif surrounding human S836 (rodent S831 and zebrafish S771). **B** Phosphorylation of S831 on AFF4 from 10-week versus 73-week-old mouse bone ($n = 3$ biological replicates/group, *$q < 0.1$; limma moderated $t$-test with Benjamini Hochberg FDR). **C**–**D** Western-blot analysis of HEK293 lysates ($n = 3$ biological replicates) for known insulin-regulated phosphorylation events in the presence of an Akt or P70S6K inhibitor. Quantification of mouse S831 phosphorylation on AFF4 by LC-MS/MS is also shown. **E** P70S6K in vitro kinase assay on purified mouse AFF4 by LC-MS/MS ($n = 2$ biological replicates/condition, *$p < 0.01$, unpaired two-sided $t$-test). **F**–**G** CRISPR/Cas9 double knockdown of $rps6kb1/2$ in zebrafish or scramble negative control (scr)(scale bar = 1 mm) followed by assessment of growth and mineralization via Alizarin red staining (***$p < 0.001$, unpaired two-sided $t$-test), (**H**) and western-blot analysis of tailfin P70S6K protein levels and phosphorylation of S235/6 on RPS6. Phosphoproteomics of CRISPR/Cas9 double knockdown ($n = 8$ biological replicates/condition; each replicant a pool of $n = 15$ individual fins) of $rps6kb1/2$ in zebrafish or scramble negative control (scr) with (**I**) volcano plot showing the phosphopeptide Log2 fold-change (r$ps6kb1$/2 versus scr) against the -Log10 $p$-value and (**J**) highlighting the phosphorylation of S771 on AFF4 and S236/40 on RPS6 (*$p < 0.05$, **$p < 0.01$, unpaired two-sided $t$-test with permutation-based FDR). Data are represented as +/− standard error of the mean. Source data are provided as a Source Data file.

following a reduction in insulin signaling[4,5] (Fig. 5A). Proteomic analysis quantified 5556 protein groups in control or IR osteoblasts and given the emphasis of our investigations is to understand the potential mechanisms of transcriptional defects, we focused on a transcription factor enrichment analysis using the ChEA3db (Supplementary Fig. 2A–C and Supplementary Data 9). Consistent with our in vivo data, IR osteoblasts displayed a significant reduction in the abundance of known target genes for the transcription factors E2F4 and FOS. These data suggest that IR osteoblasts have a reduced ability to activate IEG. We next investigated acute signaling in control or IR osteoblasts treated with or without insulin for 20 min IR resulting in attenuated insulin-induced phosphorylation of S473 on Akt along with pT389 on P70S6K and the substrate S235/6 on RPS6 (Fig. 5B–C). Targeted phosphoproteomics revealed attenuated insulin-induced phosphorylation of S831 on AFF4 under IR conditions, consistent with our annotation of this phosphosite as a P70S6K substrate (Fig. 5D and Supplementary Fig. 3). We did not observe global changes in phosphorylation of the CTD S2 or S5 on RNAP2 following insulin stimulation, suggesting no change in the activity of CyclinT1/CDK9 in the P-TEFb complex or CDK12/13, two key checkpoints in the release of paused RNAP2.

To investigate gene activation and estimate transcriptional elongation, total RNAP2 ChIP-seq was performed in control or IR osteoblasts with or without acute insulin stimulation. Immunoprecipitated RNAP2 was enriched across transcription start sites relative to negative control input consistent with the majority of RNAP2 being proximally promoter paused across the genome[15], and there was no global effect of insulin treatment in control or IR osteoblasts (Fig. 5E). We calculated the traveling ratio of each gene (amount of promotor-paused RNAP2 / the amount of RNAP2 in the gene body) to estimate the amount of RNAP2 released into productive elongation in the acute insulin-response in control vs IR osteoblasts (Supplementary Data 10). Here, lower traveling ratios are indicative of more RNAP2 released into productive elongation. Acute insulin stimulation increased the release of RNAP2 into the gene-body (observed as a decrease in traveling ratio) of several IEGs including *Egr1, Fus, Fbxl20, Errfi1*, and *Jun*, ($p < 0.05$, unpaired $t$-test) (Fig. 5F–G), findings that were previously observed with other growth factors following EGF stimulation[64]. IR generally attenuated the release of RNAP2 into productive elongation in response to acute insulin stimulation, suggesting a specific defect in transcriptional elongation. For example, under control conditions, acute insulin stimulation decreased the RNAP2 traveling ratios of *Jun*

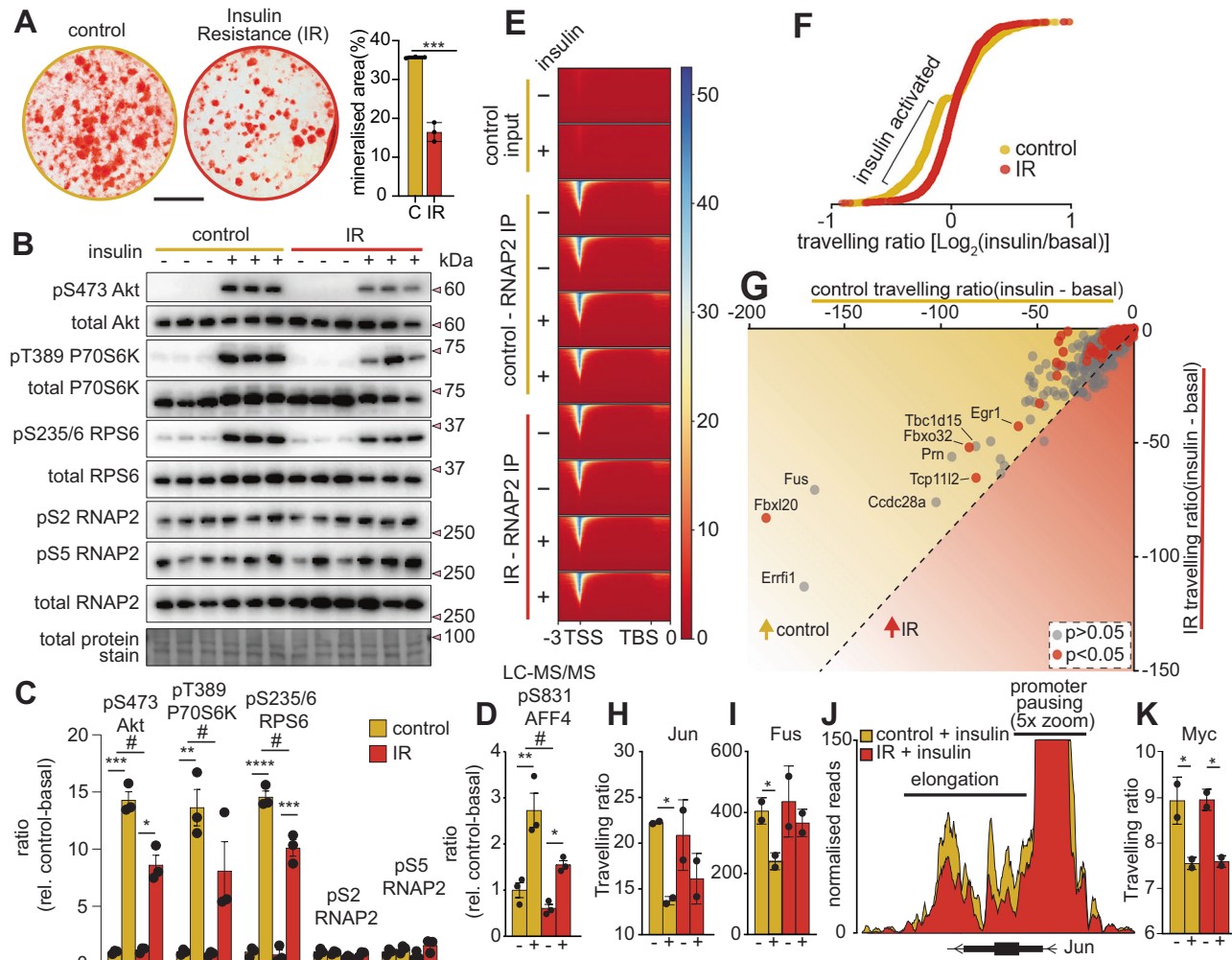

**Fig. 5 | Osteoblast insulin resistance is associated with reduced phosphorylation of S831 on AFF4 and attenuated transcriptional elongation. A** Alizarin red mineralization staining, **B** western-blot analysis, and **C** densitometry of Kusa 4B10 control or hyperinsulinemic-induced insulin-resistant (IR) osteoblasts stimulated with or without 20 min of insulin ($n = 3$ biological replicates/per condition, *$p < 0.05$, **$p < 0.01$, ***$p < 0.005$, ****$p < 0.001$; unpaired two-sided $t$-test; #$p < 0.05$, two-way ANOVA). **D** Quantification of mouse S831 phosphorylation on AFF4 by LC-MS/MS from Kusa 4B10 control or hyperinsulinemic-induced IR osteoblasts stimulated with or without 20 min of insulin ($n = 3$ biological replicates, **$p < 0.01$; unpaired two-sided $t$-test; #$p < 0.05$, two-way ANOVA). **E** ChIP-seq heatmap of RNAP2 enrichment across the transcription start site (TSS), and **F** RNAP2 traveling

ratio in Kusa 4B10 control or hyperinsulinemic-induced IR osteoblasts stimulated with or without 20 min of insulin. **G** Scatterplot of the insulin-activated genes showing the RNAP2 traveling ratio differences following 20 min of insulin stimulation of Kusa 4B10 control or hyperinsulinemic-induced IR osteoblasts with individual traveling ratio of Jun (**H**) and Fus (**I**) ($n = 2$ biological replicates/ condition, *$p < 0.05$, unpaired two-sided $t$-test). **J** ChIP-seq tracks of RNAP2 across Jun promoter and gene body from insulin stimulated Kusa 4B10 control or hyperinsulinemic-induced IR osteoblasts. **K** Individual RNAP2 traveling ratios of Myc ($n = 2$ biological replicates/ condition, *$p < 0.05$, unpaired two-sided t-test). Data are presented as +/− standard error of the mean. Source data are provided as a Source Data file.

and *Fus* by ~2-fold but this was markedly attenuated under IR conditions resulting in reduction of released and elongating RNAP2 (Fig. 5H–J). Although many IEGs displayed reduced activation under IR, several genes were not defective, such as *Myc*, which was activated in response to acute insulin stimulation to similar levels in control and IR cells (Fig. 5K). These data reveal a correlation between the phosphorylation status of S831 on AFF4 and paused-release of RNAP2, suggesting this phosphosite, at least in part, may mediate the defect in transcriptional elongation of specific genes under insulin-resistant conditions.

### Phosphorylation of S831 on AFF4 promotes gene specific insulin activation

We next investigated if phosphorylation of S831 on AFF4 plays a direct role in insulin dependent transcriptional regulation. We mutated several Ser/Thr amino acids around the S831

phosphorylation site as previous studies have shown 'hot-spot' phosphorylation which can compensate following single mutations[61]. HEK293T cells expressing FLAG-tagged AFF4-wild-type (WT) or FLAG-tagged AFF4-T829/S831/S833/S834/S835A mutant (AFF4-A-mutant) were stimulated with or without insulin for 40 min. The alanine mutations did not influence AFF4 protein abundance relative to WT levels and did not alter insulin-dependent Akt/P70S6K signaling consistent with this phosphosite being downstream of the insulin signaling pathway (Fig. 6A–B). Transcriptomic analysis identified 247 insulin-regulated genes in WT cells, and of these 150 were increased including known growth factor-regulated IEG such as *Egr1* and *Fos* which were robustly activated ($q < 0.05$, unpaired t-test with Benjamini Hochberg FDR)(Fig. 6C) (Supplementary Data 11). Cells expressing the AFF4-A-mutant generally displayed attenuated insulin-responses with 81 genes significantly different to WT (Fig. 6C). For example, *Egr1* was ~7-fold up-regulated in WT cells but

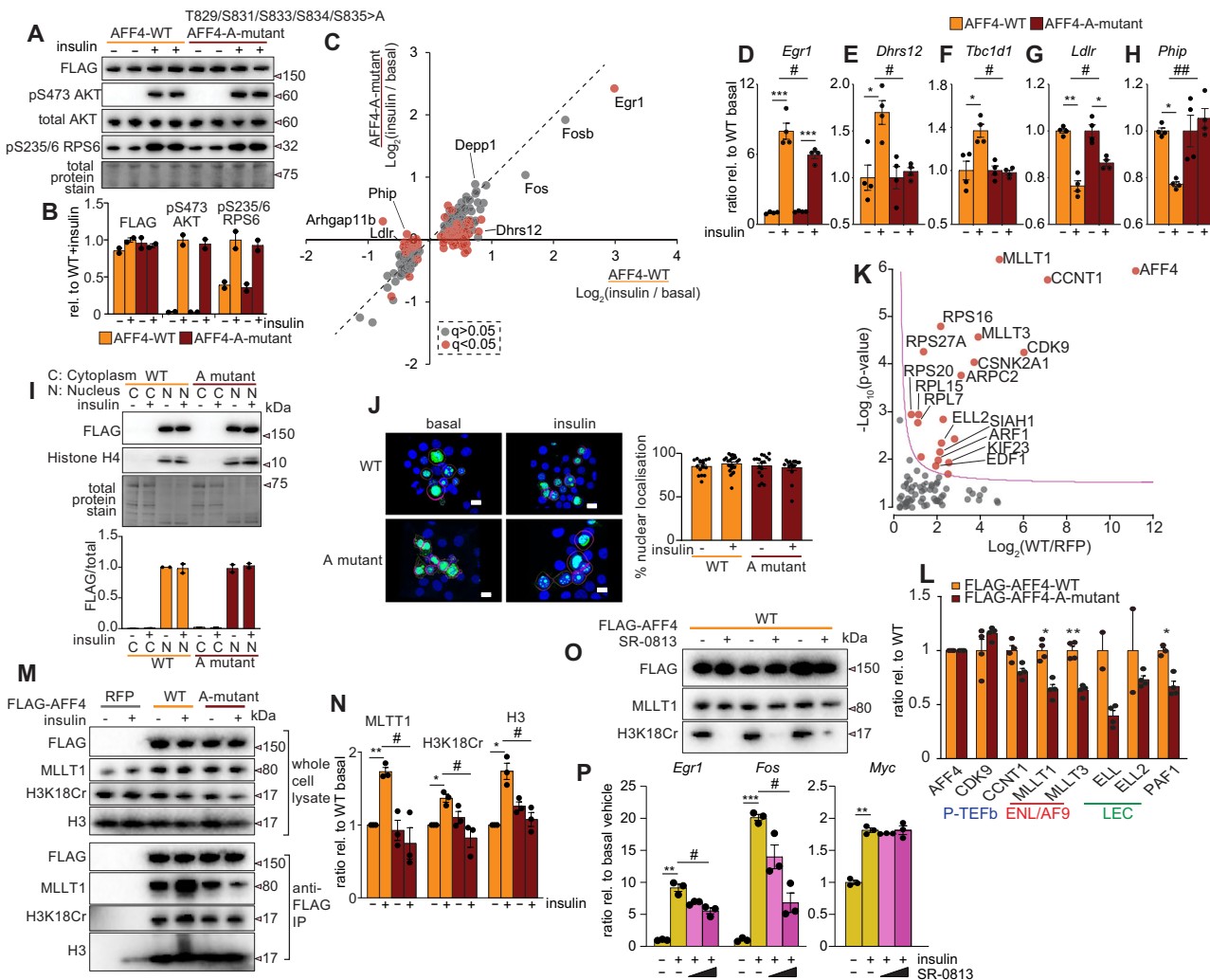

**Fig. 6 | Phosphorylation of S831 on AFF4 promotes gene specific insulin activation via ENL/AF9 recruitment to the SEC. A** Western-blot analysis, and (**B**) densitometry of HEK293T cells expressing *N*-FLAG-AFF4-wild-type (WT) or *N*-FLAG-AFF4-S829/S831/3/5/8 > A (A-mutant) stimulated with or without 20 min of insulin (*n* = 2 biological replicates/per condition). **C** Scatterplot showing insulin regulated genes and their differential response following 40 min of insulin stimulation in HEK293T cells expressing *N*-FLAG-AFF4-WT or *N*-FLAG-AFF4-S829/S831/3/5/8 > A mutant with individual genes plotted in (**D–H**) (*n* = 4 biological replicates/condition, *q < 0.05, **q < 0.01, ***q < 0.001, unpaired two-sided *t*-test with Benjamin Hochberg FDR; #p < 0.05, two-way ANOVA). **I** Western-blot analysis of nuclear versus cytoplasmic subcellular fractionation with densitometry (*n* = 2 biological replicates/condition), and (**J**) % nuclear localization assessed by immunofluorescence microscopy with anti-FLAG (green) and nuclear DAPI staining (blue). Scale bar = 10 μm. **K** Anti-FLAG immunoprecipitation and analysis of interacting proteins by LC-MS/MS from HEK293T cells expressing *N*-FLAG-AFF4-WT stimulated

with 30 min of insulin (pink line indicates *q* < 0.05, unpaired *t*-test with permutation-based FDR and S0 = 0.1), and (**L**) relative quantification of enriched proteins from cells expressing *N*-FLAG-AFF4-S829/S831/3/5/8 > A vs WT (*n* = 4 biological replicates/condition, *q < 0.05, **q < 0.01, unpaired two-sided *t*-test with Benjamin Hochberg FDR). **M** Immunoprecipitation and western-blot analysis, and (**N**) densitometry of HEK293T cells expressing *N*-FLAG-AFF4-wild-type (WT) or *N*-FLAG-AFF4-S829/S831/3/5/8 > A (A-mutant) stimulated with or without 30 min of insulin (*n* = 3 biological replicates/condition,*p < 0.05, unpaired two-sided *t*-test; #p < 0.05, two-way ANOVA). **O** Anti-FLAG immunoprecipitation of *N*-FLAG-AFF4-WT in the presence or absence of 10 μM of the YEATS domain inhibitor SR-0813. **P** qPCR of HEK293T cells treated with or without 30 min of insulin in presence of 1 μM or 10 μM of the YEATS domain inhibitor SR-0813 (*n* = 3 biological replicates, **p < 0.01, ***p < 0.001, unpaired two-sided *t*-test; #p < 0.05, two-way ANOVA). Data are presented as mean values +/− standard error of the mean. Source data are provided as a Source Data file.

only ~5-fold up-regulated in AFF4-A-mutant cells (Fig. 6D) while the oxidoreductase *Dhrs12* and the Rab-GAP regulator of GLUT4 translocation *Tbc1d1* were ~1.5-fold up-regulated in WT cells but failed to increase in the AFF4-A-mutant (Fig. 6E–F). Conversely, low-density lipoprotein receptor *Ldlr* and the PH-interacting protein *Phip* were significantly down-regulated in WT cells but not in cells expressing the A-mutant (Fig. 6G–H). The blunted insulin response in AFF4-A-mutant expressing cells was not observed for all genes. For example, *Depp1*, a critical regulator of FOXO-induced autophagy was ~1.8-fold up-regulated in both AFF4-WT and AFF4-A-mutant cells following insulin stimulation (Fig. 6C). Similarly, insulin stimulation resulted in the up-regulation of *Myc* by ~1.4-fold in both WT and A-mutant

expressing cells. We next compared the RNAP2 ChIP-seq in Kusa 4B10 osteoblasts to the RNA-seq data in HEK293T cells and identified 11 genes as insulin activated in both cell types. Importantly, the regulation of these genes was overall highly similar with genes such as *Egr1* and *Jun* showing attenuated insulin-dependent activation when phosphorylation of S831 on AFF4 was either reduced or ablated whereas genes such as *Myc* and *Atxn7l3b* displayed similar activation regardless of the phosphorylation status. These data suggest that phosphorylation of AFF4 is required for full insulin-dependent gene activation, not by global control of transcriptional elongation but by fine-tuning SEC-mediated transcriptional regulation at discrete genomic loci.

## Phosphorylation of S831 on AFF4 regulates ENL/AF9 recruitment to the SEC

We next sought to probe the molecular functions of AFF4 S831 phosphorylation and investigate how this modification may direct the activation of specific insulin-activated genes. First, we investigated if phosphorylation regulates nucleocytoplasmic shuffling. HEK293T cells expressing FLAG-tagged AFF4-WT or FLAG-tagged AFF4-A-mutant were stimulated with or without insulin for 20 min and subjected to: (i) nuclear/cytoplasmic subcellular fractionation followed by western-blotting, or (ii) immunofluorescence microscopy. We observed almost exclusive nuclear localization of both FLAG-tagged AFF4-WT and AFF4-A-mutant with no change following acute insulin stimulation (Fig. 6I–J) confirming that phosphorylation does not regulate trafficking. The S831 phosphosite on AFF4 lies within a disordered region adjacent to the ENL/AF9 binding site (Fig. 4A). We therefore hypothesized that phosphorylation may regulate protein:protein interactions within the SEC. To investigate this, HEK293T cells expressing RFP negative control, FLAG-tagged AFF4-WT or FLAG-tagged AFF4-A-mutant were stimulated with insulin for 30 min followed by immunoprecipitation and analysis of interacting proteins by LC-MS/MS. Successful enrichment of the SEC was observed including the P-TEFb complex (CDK9 and CyclinT1 (CCNT1)), the ENL/AF9 complex (MLLT1/3), and components of the Little Elongation Complex (LEC; ELL/2) and Polymerase-Associated Factor 1 (PAF1) ($q < 0.05$, $t$-test with permutation-based FDR) (Fig. 6K and Supplementary Data 12). Both WT and AFF4-A-mutant enriched the P-TEFb complex to similar levels (Fig. 6L). The enrichment of similar levels of the P-TEFb complex is consistent with our transcriptomics data, suggesting that global transcriptional elongation is not perturbed in AFF4-A-mutant cells. In contrast, there was a significant reduction in the enrichment of other SEC members, particularly MLLT1/3 of the ENL/AF9 complex and PAF in AFF4-A-mutant cells (Fig. 6L). To validate the increased association of AFF4 with the ENL/AF9 complex in response to S831 phosphorylation, and to further investigate dependence of insulin on this association, HEK293T cells expressing FLAG-tagged AFF4-WT or FLAG-tagged AFF4-A-mutant were stimulated with or without insulin for 30 min followed by immunoprecipitation and analysis by western-blotting. Insulin increased the interaction between AFF4-WT and MLLT1, but this was attenuated in the AFF4-A-mutant (Fig. 6M–N). MLLT1 is a chromatin reader that specifically binds crotonylated histones via the YEATS domain and enhances transcriptional elongation at active chromatin sites in the genome[65]. As such, we next asked if the increased association of AFF4 and MLLT1 in response to S831 phosphorylation results in enhanced recruitment to crotonylated histones. To test this, nucleosomes were prepared from HEK293T cells expressing FLAG-tagged AFF4-WT or AFF4-A-mutant treated with or without insulin stimulation. Anti-FLAG immunoprecipitation revealed an increase in the enrichment of K18 crotonylation on Histone H3 in response to insulin stimulation, and mutation of AFF4 S831 attenuated this response (Fig. 6M–N). We next hypothesized that inhibiting the YEATS domain and blocking the interaction of MLLT1/3 of the ENL/AF9 complex with crotonylated histones would attenuate insulin-dependent gene activation. To achieve this, we used the recently developed amido-imidazopyridine inhibitor of the YEATS domain, SR-0815[66]. To test inhibition, nucleosome lysates were generated from cells expressing FLAG-AFF4-WT and treated with or without SR-0815 during anti-FLAG immunoprecipitation (Fig. 6O). SR-0815 almost completely blocked the enrichment of K18 crotonylated Histone H3 confirming successful blocking of the YEATS domain. Finally, to investigate the importance of the interaction between the ENL/AF9 YEATS domain and crotonylated histone binding on the insulin-dependent gene activation, HEK293T cells were treated with or without SR-0815[66]. Acute insulin stimulation increased the activation of *Egr1* and *Fos*, and SR-0185 dose dependently inhibited this response (Fig. 6P). In contrast, the insulin-dependent activation of *Myc* was not

affected by YEATS domain inhibition. These data are consistent with our total RNAP2 ChIP-seq where insulin increased the activation of *Myc* to similar levels in both control cells and in states of low AFF4 phosphorylation induced by IR (Fig. 5K). Taken together, our data reveal that insulin-dependent phosphorylation of S831 on AFF4 recruits the chromatin readers ENL/AF9 to crotonylated histone marks resulting in the activation of specific genes.

## Discussion

Our study represents the first in vivo phosphoproteomic data of bone and characterization of insulin-dependent changes during ageing and/or IR. To prioritize candidates in this resource, we integrated the insulin-regulated phosphoproteins with BMD from human GWAS, and identified phosphosites conserved in zebrafish. These were further integrated with a machine learning analysis to predict KSRs of Akt/P70S6K and mTOR as the activation of these kinases was attenuated in aged bone. To provide evidence that a subset of these candidate phosphoproteins are causal regulators of bone, we developed a semi-high throughput in vivo functional genomic screen of bone development and mineralization in juvenile zebrafish. Six regulators were identified, with four having genetic associations to BMD in human GWAS (*EI24, RNPS1, STARD3NL* and *AFF4,*)[51].

Our study provides a blueprint of the signaling pathways responsible for the action of insulin in bone which ultimately culminates in the activation of specific genes important for cell growth, and the development and maintenance of the skeleton. Insulin-dependent phosphorylation of transcription factors can activate the transcriptional cycle, and insulin has also been shown to regulate the post-transcriptional processing of mRNA[67,68]. However, it is currently unknown whether insulin regulates other steps in mRNA production such as RNAP2 pause-release and transcriptional elongation. This is an important question because these later steps of the transcriptional cycle are emerging as the key regulatory checkpoints and rate-limiting steps of gene expression[14]. Specifically, upon initiation, RNAP2 begins transcription but stalls directly adjacent to the promoter and awaits further signals. How does insulin signaling activate the later steps of the transcriptional cycle, and furthermore, how does it achieve this in a gene specific manner? More importantly, are these signals perturbed under IR and/or ageing conditions and a contributing factor to aberrant gene expression? Our phosphoproteomic analysis identified insulin-dependent phosphorylation of AFF4 that was attenuated in the bones of aged and IR mice. We show that AFF4 is required for skeletal development in zebrafish, and phosphorylation of S831 is mediated by P70S6K which is defective in IR osteoblasts. Reduced phosphorylation of AFF4 was associated with a decrease in the release of paused RNAP2 into productive elongation, and this was not a global effect, but was restricted to precise locations in the genome. This specificity of action was reflected by selective enhancement of the expression of specific IEGs by insulin-dependent phosphorylation of S831 on AFF4, and recruitment of the ENL/AF9 complex to the SEC. ENL/AF9 are chromatin readers that specifically bind crotonylated histones via their YEATS domain and enhance transcriptional elongation at active chromatin sites in the genome[65]. Hence, we propose the following mechanism for insulin-mediated gene activation (Fig. 7). Under low insulin steady-state conditions, basal transcriptional initiation complexes have low activity, and RNAP2 is paused proximal to the promoter. Upon insulin stimulation, transcriptional initiation complexes are rewired via several kinases acting on transcription factors such as Akt/FOXO, GSK3/FOXK1, JNK/AP1 and ERK/ELK1, amongst others. ERK and P-TEFb corporately phosphorylate the CTD of RNAP2, NELF and DSIF to facilitate pause-release of RNAP2[18]. At the same time, P70S6K is activated in the nucleus[69], and phosphorylates S831 on AFF4 which recruits ENL/AF9 to the SEC and active crotonylated chromatin. These crotonylation sites are located at precise regions of the genome[65] resulting in enhanced expression of specific genes. Under IR

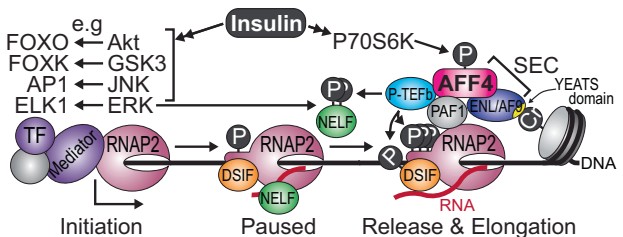

**Fig. 7 | Model of insulin-dependent transcriptional initiation, pause, release and elongation.** The diagram highlights the central role of AFF4 phosphorylation and regulation of the Super Elongation Complex (SEC).

conditions, the reduced activity of several kinases attenuate transcriptional initiation and diminished P70S6K activity attenuates ENL/AF9 recruitment to the SEC and gene expression.

While our focus in this study was AFF4, future experiments characterizing the potential regulatory role of EI24, RNPS1 and STARD3NL phosphorylation on protein function are warranted. For example, in the case of EI24, the magnitude of the insulin response of S326/30 phosphorylation was significantly attenuated in the bones of 73-week-old mice and is conserved in zebrafish with an mTOR predictions score of 0.86 (range 0–1). These phosphosites sit in a C-terminal hotspot of post-translational modifications adjacent to ubiquitinylation of K323 and monomethylation of K335[70] suggesting possible regulatory crosstalk based on their proximity[71]. In the case of RNPS1, its conserved insulin-dependent S155/157 phosphorylation site was significantly increased in bones for 10-week-old mice but not in aged mice and has an mTOR prediction score of 0.85. These phosphosites sit directly adjacent to a conserved RRM RNA-binding domain which forms a characteristic alpha/beta sandwich fold suggesting phosphorylation may modulate RNA-binding. Finally, the conserved S39 phosphorylation site on STARD3NL was down-regulated following insulin stimulation in the bones of 73-week-old mice and lies within an alpha-helix on the cytoplasmic side of the transmembrane MENTAL domain which is required for targeting this protein to the late endosome[72]. Hence, phosphorylation may regulate endosome trafficking which is important because STARD3NL is required for the formation of organelle contact sites at the ER and contains a cholesterol binding domain implicating this protein in lipid transfer[73]. Further work on these phosphoproteins, and their role in the skeleton is warranted.

Our data reveal a previously unexplored function of P70S6K in regulating transcriptional elongation, expanding its role beyond its actions on protein synthesis, cell survival and skeletal development[74]. Given that the S831 phosphorylation site on AFF4 is evolutionarily conserved across the animalia kingdom, this mechanism is likely relevant for insulin-mediated gene-selective activation across many species. This is exciting because reducing insulin signaling extends longevity in flies[75], worms[76], and mice[77], and the species conservation of this pathway means such findings may be applicable to humans. Recent studies have shown that ageing is associated with changes in RNAP2 pausing and transcriptional elongation. Specifically, Debès and co-workers showed that transcriptional elongation rates across several species are increased with age, and reducing insulin signaling can decrease elongation rates and extended lifespan[78]. Gyenis and co-workers showed that ageing leads to an increase in RNAP2 stalling and a decrease in nascent RNA synthesis[79]. While these studies appear to be contradictory, further analysis has shown that the results are complementary with ageing associated with increased transcriptional activity compensating for stalled RNAP2 and non-productive RNA synthesis[80]. It is tantalizing to speculate that age-associated changes in the phosphorylation of AFF4 by P70S6K is a key mechanistic link between insulin signaling, transcriptional elongation and longevity.

Further supporting this hypothesis, female mice lacking the short isoform of P70S6K (S6K1) have increased longevity[81]. It is important to note that the previously identified age-associated changes in transcriptional elongation and RNAP2 stalling are mainly affecting long genes and early termination at intronic polyA sites. However, we show that P70S6K-dependent phosphorylation of AFF4 is mainly regulating shorter IEGs in response to acute insulin stimulation. Future studies investigating the role of P70S6K-dependent phosphorylation of AFF4 on RNAP2 stalling during ageing will further clarify its role in longevity.

## Limitations of the study

Our phosphoproteomic analysis of mouse bone was performed only in male mice which is an important limitation given previously identified sex signaling differences in stem cells derived from diabetic subjects[82], and the known differences in bone structure/remodeling between male and females[83]. Mice were injected with relatively high levels of insulin via intraperitoneal injection which, although it did not induce lethargic signs of hypoglycemia, may not represent the physiological insulin response following a meal. For the analysis of the in vivo phosphoproteomics of bone, we chose to use the Benjamini-Hochberg method to adjust for multiple hypothesis testing which is more stringent than other methods to adjust $p$-values (e.g., permutation-based FDR) but decided to use a less stringent cut-off of $q < 0.1$ to capture trends as performed by previous in vivo proteomic studies[84,85]. Our functional genomic screen in zebrafish injected Cas9/4gRNA RNPs into zebrafish embryos resulted in mosaic editing throughout the entire animal. CRISPR editing was only confirmed in positive hits of the screen via PCR. Of the positive hits, we only generated germline mutants of *aff4* via backcrossing and precision mapping of edits via Sanger sequencing. It is also important to note that kinases other than P70S6K may phosphorylate S831 on AFF4.

## Methods
### Animals

All mouse experiments were approved by The University of Melbourne Animal Ethics Committee (AEC ID1914940) and conformed to the Australian code for the care and use of animals for scientific purposes as stipulated by the National Health and Medical Research Council of Australia. C57BL/6 J mice (JAX 000664) were obtained from Animal Resource Center (WA, Australia). Mice were housed at 22 °C ( + /−1 °C) in groups of five/cage and maintained on a standard chow diet (Specialty Feeds, Australia) with a 12 h light/dark cycle and *ad libitum* access to food and water.

All the zebrafish experiments were approved by The University of Melbourne Animal Ethics Committee (2021-22235-23572-4 and 2023-10473-38935-6) and conformed to the National Health and Medical Research Council of Australia guidelines regarding the care and use of experimental animals. Wildtype AB zebrafish were maintained under a 14 h light/10 h dark cycle at the *Danio rerio* University of Melbourne (DrUM) facility. For routine breeding and microinjection experiments, mating pairs were set up in breeder boxes (Tecniplast, Australia) on the evening prior, separated by a divider for timed mating. When required, the dividers were removed to allow embryo spawning for 10–15 min. All generated embryos were maintained at 28.5 °C in E3 embryo medium (5 mM NaCl, 0.17 mM KCl, 0.33 mM $CaCl_2$, 0.33 mM $MgSO_4.7H_2O$ in RO water) until 5 days post fertilization (dpf), after which they were transferred to an online freshwater/RO circulatory system in 3.5 L tanks at 10 fish/L density (Tecniplast, Italy) and raised on a paramecium diet until 15 dpf, live *Artemia* and brine diet until 60 dpf and finally on a marine dried food and live *Artemia* diet throughout adulthood. For alizarin red staining experiments, the fish were fasted for 12 h prior to euthanasia to clear gut contents. Zebrafish were euthanized by complete immersion in 1000 mg/L tricaine methanesulfonate (MS-222, Sigma Aldrich, Australia) until opercula movements

and scoot/swim behavior had stopped and used in experiments as described below.

## Cell lines

Female HEK293 cells were purchased from ATCC (ID: CRL-1573) and grown in Dulbecco's Modified Eagle Medium (DMEM) (GIBCO by Life Technologies; # 11995065), supplemented with 10% fetal bovine serum (FBS) (Life Technologies; #26140079), pyruvate and GlutaMAX (GIBCO by Life Technologies). Female Kusa 4B10 stromal stem cells were grown in Minimum Essential Medium alpha (MEMalpha) (GIBCO by Life Technologies; #12561056) supplemented with 10% fetal bovine serum (FBS) (Life Technologies; #26140079), pyruvate and GlutaMAX (GIBCO by Life Technologies). Cells were kept at 37 °C and 5% $CO_2$ in a humidified incubator Direct Heat $CO_2$ Incubator featuring Oxygen Control (In Vitro Technologies). All cell stocks were regularly checked for absence of mycoplasma with the Mycoplasma Detection Kit (Jena Bioscience; # PP-401).

## Mouse body composition and glucose/insulin measurements

Metabolic measurements were undertaken in the Melbourne Mouse Metabolic Phenotyping Platform (The University of Melbourne, Australia). Body composition analysis to determine whole-body adiposity and lean mass was measured using TD-NMR mini spec (Bruker Optics Inc., Billerica, MA). Fasted (12 h fast overnight) plasma insulin or glucose levels were determined using a Rat/Mouse Insulin ELISA (EZRMI-13K, Merck Millipore, CA) or an Accu-Check glucometer, respectively. The HOMA-IR was calculated using the equation [(glucose × insulin) / 405].

## Microcomputed tomography of mouse bone

Tibiae were fixed in 4% paraformaldehyde for 24 h, rinsed in phosphate-buffered saline (PBS) and stored in 70% ethanol. Tibiae were scanned in ethanol using the Skyscan 1276 micro-CT (Bruker, Kontich, Belgium) at 9 μm resolution, 0.25 mm aluminum filter, 56 kV voltage, 200 μA, 560 ms exposure time, 0.4 ° step rotation with frame averaging of 2. Images were reconstructed using NRecon (v1.7.4.6, Bruker), prepared using Dataviewer (v1.5.6.2, Bruker) and analyzed using CTAnalyzer (CTAn; v1.18.8.0, Bruker). Representative reconstructions were visualized in CTVox (v3.3.0 r1403, Bruker) using thresholded datasets or a custom pseudodensity filter on raw datasets. Trabecular bone analysis was performed on the proximal metaphysis, where the region for analysis commenced at 0.5 mm distal of the growth plate and extended distally for a total of 2 mm. Cortical bone analysis was performed on a region commencing at 7.5 mm from the top of the tibia and extended distally for 2 mm. The thresholds used for trabecular and cortical bone were 0.47 g/cm3 and 0.87 g/cm3 calcium hydroxyapatite, respectively.

## Proteome and phosphoproteome sample preparation of mouse bone

Tibia bones cleared of bone marrow cells were ground into powder under liquid nitrogen, lysed in 6 M guanidine HCl (Sigma; #G4505), 100 mM Tris pH 8.5 containing 10 mM tris(2-carboxyethyl)phosphine (TCEP) (Sigma; #75259) and 40 mM 2-chloroacetamide (CAA) (Sigma; #22790) by tip-probe sonication (QSonica). The lysate was heated at 95 °C for 5 min and centrifuged at 20,000 × g for 10 min at 4 °C. The supernatant was diluted 1:1 with water and precipitated overnight with 5 volumes of acetone at −20 °C. The lysate was centrifuged at 4,000 x g for 5 min at 4 °C and the protein pellet was washed with 80% acetone. The lysate was centrifuged at 4000 × g for 5 min at 4 °C and the protein pellet was resuspended in Digestion Buffer (10% 2,2,2-Trifluoroethanol (Sigma; #96924) in 100 mM HEPEs pH 8.5). Protein was quantified with Pierce™ BCA (ThermoFisher Scientific) and 300 μg of protein was normalized to a final volume of 80 μl in Digestion Buffer, and digested with sequencing grade trypsin (Sigma; #T6567) and sequencing grade

Lys-C (Wako; #129-02541) at a 1:50 enzyme:substrate ratio overnight at 37 °C with shaking at 2000 x rpm. Eight hundred microgram of tandem mass tags (TMT) was resuspendended in 40 μl of acetonitrile and added directly to the digested peptides. The TMT labeling scheme and all raw data have been uploaded to the ProteomeXchange Consortium PRIDE repository (see Data Availability section). The reaction was incubated at room temperature for 1.5 h, deacylated with 0.3% (w/v) of hydroxylamine for 10 min at room temperature and quenched to a final volume of 1% trifluoroacetic acid (TFA). The TMT-labeled peptides were pooled and dried to ~250 ul in a vacuum centrifuge. Phospho-peptides were enriched using a modified EasyPhos protocol[86]. Briefly, samples were diluted to a final concentration of 50% isopropanol containing 5% TFA and 0.8 mM $KH_2PO_4$ and incubated with 30 mg of $TiO_2$ beads (GL Sciences; #5010-21315) for 8 min at 40 °C with shaking at 2000 x rpm. The beads were washed four times with 60% isopropanol containing 5% TFA and resuspended in 60% isopropanol containing 0.1% TFA. The bead slurry was transferred to in-house packed C8 microcolumns (3 M Empore; #11913614) and phosphopeptides eluted with 40% acetonitrile containing 5% ammonium hydroxide. Peptides were acidified to a final concentration of 1% TFA in 90% isopropanol and purified by in-house packed SDB-RPS (Sigma; #66886-U) microcolumns. The purified peptides were resuspended in 2% acetonitrile in 0.1% TFA and stored at −80 °C prior to offline fractionation using neutral phase C18BEH HPLC as previously described[87].

## Zebrafish caudal fin phosphoproteomic and proteomic sample preparation

For phosphoproteomics analysis of P70S6K knockdown, 21 dpf gRNA *rps6kb1a/b* or gRNA scramble negative control F0 crispants (*n* = 105) were fasted for 16 h overnight prior to tissue collection. Post fasting, the fish were allowed feeding for 30 min and then immediately euthanized in MS-grade water (as described above) for tissue collection. For proteomic analysis of AFF4 knockdown, 35 dpf gRNA *aff4* or gRNA scramble negative control F0 crispants (*n* = 14) were fasted for 16 h overnight prior to tissue collection. The whole caudal fin from each fish was rapidly dissected on ice, collected in 1.5 ml protein LoBind tubes (Eppendorf, Germany) and snap frozen on dry ice. Dissected caudal fins (*n* = 8 replicates/condition; each replicant contained a pool of *n* = 15 individual fins) were lysed in w/v 2% SDS in 100 mM Tris, pH 8.5 lysis buffer by tip probe sonication (QSonica) at 20% power setting for 10 s followed by heat denaturation at 95 °C for 5 min and centrifuged at 18,000 g for 15 min at 4 °C to pellet cellular debris and separate out the top lipid layer. An additional 30 caudal fins from non-injected AB wild-type zebrafish were also lysed in parallel to be used in the generation of phosphopeptide spectral library. Care was taken to avoid lipid contamination while retrieving the protein supernatant for downstream processing. Sample protein concentration was estimated using the Pierce™ BCA protein assay kit (ThermoFisher Scientific, USA) and phosphoproteomic samples of P70S6K knockdown were normalized to 2.5 μg/μl (total 1 mg) protein while proteomic samples of AFF4 knockdown were normalized to 66.7 ng/μl (total 6 μg) in lysis buffer. Normalized protein samples were subjected to reduction and alkylation in a single-pot reaction at 45 °C for 5 min using 10 mM TCEP/ 40 mM 2-CAA in 100 mM Tris, pH 8.5 buffer. The samples were then subjected to the single-pot, solid-phase-enhanced sample-preparation (SP3) method to remove SDS using Sera-Mag SpeedBead Magnetic Carboxylate hydrophilic and hydrophobic beads in a total bead:protein ratio of 10:1 as previously described[88]. Cleaned-up samples were resuspended in 10% trifluoroethanol in 100 mM Tris, pH 8.5 digestion buffer and protein digestion was performed with sequencing grade trypsin (Promega) and Lys-C (Wako, Japan) enzyme (protein:enzyme ratio 100:1) for 16 h at 37 °C with shaking at 1600 rpm. For phospho-proteomic analysis, the digested peptides were then subjected to phosphopeptide enrichment using $TiO_2$ beads (beads:protein ratio 12:1) as previously described[86] and eluted in the EasyPhos elution

buffer at 2000 g from in-house C8 stage tips. The eluted phospho-peptides from P70S6K knockdown or the peptides from AFF4 knock-down were subjected to further clean-up using in-house SDP-RPS stage tips and eluted with v/v 80% acetonitrile/5% NH₄OH at 500 g for 5 min. Eluted peptides were dried in a vacuum concentrator (Eppendorf, Germany) at 45 °C for 45 min and the dried peptides were resuspended in 2% acetonitrile/0.1% TFA. Phosphopeptides enriched from the wild-type AB caudal fins were further fractionated as described above for spectral library generation while the r*ps6kb1a/b* or gRNA scramble negative control were analyzed directly by LC-MS/MS.

## Immunoprecipitation of AFF4 for interacting protein analysis

Mouse Aff4 cDNA (NM_033565.2) was purchased from VectorBuilder with an N-terminal 3xFLAG tag as either wild-type (WT) or T829A/S831A/S833A/S834A/S835A mutations (A-mutant) containing CMV promoter cloned into a pcDNA3.1 vector. A CMV driven RFP vector in pcDNA3.1 was used as a negative control. RFP, WT and A-mutant vectors were transiently transfected at 60% confluence in a 6-well plate with 3 μg DNA, 9 μl of P3000 reagent and 9 μl of Lipofectamine 3000 prepared in reduced-serum Minimal Essential Medium (Opti-MEM) (Life Technologies). After 24 h, the media was replaced with DMEM, supplemented with 10% FBS, pyruvate and GlutaMAX and cultured for a further 2 days. On the day of harvest, cells were withdrawn from serum by incubating in DMEM, Pyruvate, GlutaMAX containing 0.2% BSA for 2 h to achieve a basal state. Relevant cells were then stimulated for 30 min with 100 nM of insulin. Cells were washed twice with cold PBS, then lysed in cold modified RIPA Buffer (10% glycerol, 150 mM NaCl, 1% NP40, 0.5% NaDOC, 0.1% SDS, 50 mM Tris pH 7.4, containing 10 mM sodium fluoride, 10 mM sodium pyrophosphate, 10 mM gly-cerophosphate, 2 mM sodium orthovanadate and EDTA-free protease inhibitor cocktail (PIC)(Roche; #11836170001)). The whole cell lysates were pushed through a 22 G needle 6 times followed by a 27 G needle 3 times followed by the addition of 250U of benzonase (Sigma; E1014) and incubated on ice for 1 h. For experiments involving the generation of nucleosomes and analysis of histone crotonlylation interaction, cells were lysed and nuclei enriched in Crude Nuclear Buffer (0.1% NP40 in PBS containing 10 mM sodium fluoride, 10 mM sodium pyropho-sphate, 10 mM glycerophosphate, 2 mM sodium orthovanadate, 1 mM sodium butyrate, 5 mM nicotinamide and EDTA-free PIC). The lysate was spun at 16,000 × $g$ for 2 min at 4 °C and supernatant discarded. The pellet containing crude nuclei were washed once with Crude Nuclear Buffer and resuspended in MNase Digestion Buffer (10 mM Tris-HCl pH 7.4, 15 mM NaCl, 60 mM KCl containing 10 mM sodium fluoride, 10 mM sodium pyrophosphate, 10 mM glycerophosphate, 2 mM sodium orthovanadate, 1 mM sodium butyrate, 5 mM nicotina-mide and EDTA-free PIC). The crude nuclei were mixed gently, cen-trifuged at 5,000 x g for 5 min at 4 °C and supernatant discarded. Crude nuclei were resuspended in MNase digestion Buffer containing 5 mM calcium chloride and treated with 4000U of Micrococcal Nuclease (New England Biolabs; M0247S) for 5 min at 37 °C. The reaction was stopped with a final concentration of 10 mM EGTA and Triton-X100 added to a final concentration of 0.1% followed by brief mixing. The whole cell lysates and the crude nucleosome lysates were spun down at 18,000 × $g$ for 10 min at 4 °C and the soluble supernatant quantified with Pierce™ BCA (ThermoFisher Scientific). During the serum withdrawal period, 2 μl of mouse anti-FLAG M2 (Sigma; #F1804) was added to 40 μl of protein-G beads (Invitrogen; #10004D) and incubated for 2 h with rotation at 4 °C. The anti-FLAG:bead complexes were washed twice with cold modified RIPA Buffer and the normalized whole cell protein lysates or the crude nucleosome lysates added to the beads followed by incubation for 2 h with rotation at 4 °C. The lysate:anti-FLAG:bead complexes were washed three times with cold modified RIPA and enriched proteins eluted with 1% SDS, 10 mM TCEP, 40 mM CAA and heat at 9 5 °C for 5 min with shaking at 1800 rpm. The samples were then either subjected to SP3 purification for AP-MS

experiments or diluted with Laemlli Buffer for western-blot analysis. For SP3, protein bound and washed beads were resuspended in 10% trifluoroethanol in 100 mM Tris, pH 7.5 digestion buffer and protein digestion was performed with 0.4 μg of sequencing grade trypsin (Promega) and 0.4 μg of Lys-C (Wako, Japan) enzyme for 16 h at 37 °C with shaking at 1600 rpm. The digested peptides were then purified using in-house SDP-RPS stage tips and eluted with v/v 80% acetonitrile/5% NH₄OH at 500 g for 5 min. Eluted peptides were dried in a vacuum concentrator (Eppendorf, Germany) at 45 °C for 45 min and the dried peptides were resuspended in 2% acetonitrile/0.1% TFA and stored at −80 °C prior to direct injection by LC-MS/MS.

## Immunoprecipitation of AFF4 for phosphosite mapping, Akt/P70S6K inhibitor analysis and P70S6K in vitro kinase assay

Wild-type Aff4 cDNA N-terminally FLAG tagged vector was transfected as described above. On the day of harvest, cells were withdrawn from serum by incubating in DMEM, Pyruvate, GlutaMAX containing 0.2% BSA for 2 h to achieve a basal state. After 1.5 h, relevant cells were treated with 10 μM of the Akt inhibitor MK2206 (Sapphire Bioscience; SYN-1162-M005) or 1 μM of the P70S6K inhibitor LY2584702 (tosylate) (Sapphire Bioscience; 15320) for 30 min. Cells were left unstimulated or stimulated with 100 nM insulin for 20 min. Cells were washed twice with cold PBS, then lysed in cold modified RIPA Buffer followed by anti-FLAG immunoprecipitation as described above. Following washing of the beads three times with modified RIPA, AFF4 was competitively eluted with 16 μg excess 3XFLAG peptide in Tris-Buffered Saline for 1 h at 4 °C with rotation. For the P70S6K in vitro kinase assay, anti-FLAG immunoprecipitated AFF4 was diluted 1:1 with 2X Kinase Buffer (50 mM Tris pH 7.4, 800 μM ATP, 20 mM MgCl₂), aliquoted in two equal amounts and treated with either MilliQ water or 250 ng of active recombinant P70S6K (Sigma; SRP0365) and incubated at 30 °C for 1 h. The anti-FLAG immunoprecipitated AFF4 from inhibitor/insulin sti-mulated cells or treated with or without recombinant P70S6K were mixed with Laemmli Buffer, heated at 65 °C for 10 min and separated on NuPAGE 4–12% Bis-Tris protein gels (ThermoFisher Scientific) in MOPS SDS Running Buffer at 160 V for 1 h at room temperature. The gel was incubated overnight in SyproRuby, imaged on a ChemiDoc (BioRad) and the AFF4 band at ~160 kDa was excised. Gel bands were washed twice in 50% acetonitrile in 50 mM ammonium bicarbonate pH 7.9 for 15 min, dehydrated with 100% acetonitrile and then reduced with 10 mM TCEP, 40 mM CAA in 50 mM ammonium bicarbonate, pH 7.9 and heated at 55 °C for 30 min. The gel bands were dehydrated with 100% acetonitrile and then rehydrated with sequencing grade trypsin (14 ng/μl) on ice for 1 h. Excess trypsin was removed and gel pieces incubated in 50 mM ammonium bicarbonate overnight at 37 °C. Pep-tides were purified using in-house SDP-RPS stage tips and eluted with v/v 80% acetonitrile/5% NH₄OH at 500 g for 5 min. Eluted peptides were dried in a vacuum concentrator (Eppendorf, Germany) at 45 °C for 45 min and the dried peptides were resuspended in 2% acetonitrile/0.1% TFA and stored at −80 °C prior to direct injection by LC-MS/MS.

## Targeted phosphoproteomics of AFF4 phosphorylation in insulin-resistant Kusa 4B10 osteoblasts

Kusa 4B10 stromal stem cells (SSCs) were maintained in MEM α sup-plemented with 10% FBS, pyruvate and GlutaMAX and differentiated into osteoblasts over 21 days with 50 μg/ml of ascorbate and 10 mM of beta-glycerophosphate. Hyperinsulinemia-induced insulin resistance was established by treating cells with 20 nM of insulin 3 times a day (08:00, 12:30 and 17:30) for 2 days. The next day, cells were withdrawn from serum by incubating in MEM α, Pyruvate, GlutaMAX containing 0.2% BSA for 2 h to achieve a basal state. Cells were left unstimulated or stimulated with 10 nM insulin for 20 min. Cells were washed twice with cold PBS, then lysed in cold 6 M guanidine HCl, 100 mM Tris pH 8.5 containing 10 mM tris(2-carboxyethyl)phosphine and 40 mM 2-chloroacetamide (Sigma; #22790) by tip-probe sonication (QSonica).

The lysate was heated at 95 °C for 5 min and centrifuged at 20,000 × *g* for 10 min at 4 °C. The supernatant was diluted 1:1 with water and precipitated overnight with 5 volumes of acetone at −20 °C. The lysate was centrifuged at 4,000 x g for 5 min at 4 °C and the protein pellet was washed with 80% acetone. The lysate was centrifuged at 4,000 x g for 5 min at 4 °C and the protein pellet was resuspended in Digestion Buffer (10% 2,2,2-Trifluoroethanol in 100 mM HEPEs pH 7.5). Protein was quantified with Pierce™ BCA (ThermoFisher Scientific) and 600 μg of protein was normalized to a final volume of 200 μl in Digestion Buffer for targeted phosphoproteomics, and digested with sequencing grade trypsin (Sigma; #T6567) and sequencing grade Lys-C (Wako; #129-02541) at a 1:100 enzyme:substrate ratio overnight at 37 °C with shaking at 2000 x rpm. Phosphopeptides were enriched using the EasyPhos protocol described above using 7.2 mg of TiO$_2$ beads. Peptides were purified using in-house SDP-RPS stage tips and eluted with v/v 80% acetonitrile/5% NH$_4$OH at 500 g for 5 min. Eluted peptides were dried in a vacuum concentrator (Eppendorf, Germany) at 45 °C for 45 min and the dried peptides were resuspended in 2% acetonitrile/ 0.1% TFA and stored at −80 °C prior to direct injection by LC-MS/MS.

## Proteome analysis of insulin-resistant Kusa 4B10 osteoblasts

Kusa 4B10 stromal stem cells (SSCs) were differentiated into osteoblasts over 21 days and either left untreated as controls or made insulin-resistant over 2 days as described above. Cells were washed twice with cold PBS and scraped in 4% sodium deoxycholate (SDC) in 100 mM Tris pH 8.5 and stored at −20 °C for total proteomic analysis. Proteins were reduced and alkylated and purified by SP3 as described above for the preparation of proteins from zebrafish caudal fins. The SP3 beads were resuspended in 10% trifluoroethanol in 100 mM Tris, pH 8.5 digestion buffer and protein digestion was performed with sequencing grade trypsin (Promega) and Lys-C (Wako, Japan) enzyme (protein:enzyme ratio 100:1) for 16 h at 37 °C with shaking at 1600 rpm. Peptides were purified using in-house SDP-RPS stage tips and eluted with v/v 80% acetonitrile/5% NH$_4$OH at 500 g for 5 min. Eluted peptides were dried in a vacuum concentrator (Eppendorf, Germany) at 45 °C for 45 min and the dried peptides were resuspended in 2% acetonitrile/0.1% TFA.

## LC-MS/MS acquisition

**Mouse bone.** Phosphopeptides and non-enriched peptides were analyzed on a Dionex 3500 nanoHPLC coupled to an Orbitrap Eclipse mass spectrometer (ThermoFisher Scientific) via electrospray ionization in positive mode with 1.9 kV at 275 °C and RF set to 30%. Separation was achieved on a 50 cm × 75 μm column packed with C18AQ (1.9 μm; Dr Maisch, Ammerbuch, Germany) (PepSep, Marslev, Denmark) over 60 min (fractionated phosphoproteomics) or 120 min (fractionated proteomics) at a flow rate of 300 nL/min. The peptides were eluted over a linear gradient of 3–23% Buffer B (Buffer A: 0.1% formic acid; Buffer B: 80% v/v acetonitrile, 0.1% v/v FA) and the column was maintained at 50 °C. The instrument was operated in data-dependent acquisition (DDA) mode with an MS1 spectrum acquired over the mass range 350–1550 m/z (120,000 resolution, 1 × 106 automatic gain control (AGC) and 50 ms maximum injection time) followed by MS/MS analysis with fixed cycle time of 3 s via HCD fragmentation mode and detection in the orbitrap (50,000 resolution, 1 × 10⁵ AGC, 150 ms maximum injection time (for phosphopeptides) or 86 ms maximum injection time (for non-labeled peptides), and 0.7 m/z isolation width). Only ions with charge state 2–7 triggered MS/MS with peptide monoisotopic precursor selection and dynamic exclusion enabled for 60 s at 10 ppm.

**Zebrafish caudal fin for analysis of P7OS6K knockdown.** Phospho-peptides were analyzed on a Dionex 3500 nanoHPLC coupled to an Orbitrap Exploris 480 mass spectrometer (ThermoFisher Scientific) via electrospray ionization in positive mode with 1.9 kV at 275 °C and RF

set to 30%. Separation was achieved on a 50 cm × 75 μm column packed with C18AQ (1.9 μm; Dr Maisch, Ammerbuch, Germany) (PepSep, Marslev, Denmark) over 60 min at a flow rate of 300 nL/min. The peptides were eluted over a linear gradient of 3–23% Buffer B (Buffer A: 0.1% formic acid; Buffer B: 80% v/v acetonitrile, 0.1% v/v FA) and the column was maintained at 50 °C. For spectral library generation using the wild-type AB zebrafish causal fins, the instrument was operated in data-dependent acquisition (DDA) mode with an MS1 spectrum acquired over the mass range 300–1600 m/z (120,000 resolution, 1 × 106 AGC and 50 ms maximum injection time) followed by MS/MS analysis with fixed cycle time of 3 s via HCD fragmentation mode and detection in the orbitrap (15,000 resolution, 1 × 10⁵ AGC, 54 ms maximum injection time, and 1.2 m/z isolation width). Only ions with charge state 2–6 triggered MS/MS with peptide monoisotopic precursor selection and dynamic exclusion enabled for 30 s at 10 ppm. For quantification of phosphopeptides enriched from gRNA scramble and gRNA *rps6kb1a/b* injected causal fins, the instrument was operated in hybrid parallel reaction monitoring and data-independent acquisition (PRM/DIA) mode with an MS1 spectrum acquired over the mass range MS1 spectrum acquired over the mass range 350–1400 m/z (60,000 resolution, 1 × 106 AGC and 50 ms maximum injection time) followed by MS/MS analysis of 50 × 13.7 m/z isolation windows over the mass range of 360.5–1033.5 m/z via HCD fragmentation mode and detection in the orbitrap (30,000 resolution, 1 × 10⁶ AGC, 55 ms maximum injection time; 30% normalized collision energy). Four additional scans were acquired per duty cycle targeting the zebrafish AFF4 S771 phos-phopeptides identified in the spectral library (TLSpQSpSSTVPSK, 691.28603 m/z, [M + 2H]2 + ; TLSpQSSSTVPSK, 651.302865 *m/z*, [M + 2H]2 + ; RTLSpQSpSSTVPSK, 729.35342 *m/z*, [M + 2H]2 + ; RTLSpQSSSTVPSK, 769.336586 *m/z*, [M + 2H]2 + ) via HCD fragmenta-tion mode and detection in the orbitrap (30,000 resolution, 1 × 10⁵ AGC, 84 ms maximum injection time, and 2.0 m/z isolation width).

**Zebrafish caudal fin for analysis of AFF4 knockdown.** *P*eptides were analyzed on a Vanquish Neo UHPLC (ThermoFisher Scientific) coupled to an Orbitrap Astral mass spectrometer (ThermoFisher Scientific), via electrospray ionization in positive mode with 1.9 kV at 275 °C and RF set to 50%. Separation was achieved on a 50 cm μPAC Neo nanoLC pillar array column (pillar dimensions 18 × 5 μm) (ThermoFisher Sci-entific) over 30 min at a flow rate of 750 nL/min. The peptides were eluted over a linear gradient of 3–23% Buffer B (Buffer A: 0.1% formic acid; Buffer B: 80% v/v acetonitrile, 0.1% v/v FA) and the column was maintained at 50 °C. The instrument was operated in data-independent acquisition (DIA) mode with an MS1 spectrum acquired over the mass range MS1 spectrum acquired over the mass range 380–980 m/z (120,000 resolution, 500% AGC and 50 ms maximum injection time) followed by MS/MS analysis of 300 × 2 *m/z* isolation windows via HCD fragmentation mode and detection in the Astral (5 × 10⁴ AGC, 3 ms maximum injection time; 27% normalized collision energy).

**Immunoprecipitated AFF4 for phosphosite mapping, Akt/P7OS6K inhibitor analysis and P7OS6K in vitro kinase assay.** Peptides were analyzed on a Dionex 3500 nanoHPLC coupled to an Orbitrap Eclipse mass spectrometer (ThermoFischer Scientific) via electrospray ioni-zation in positive mode with 1.9 kV at 275 °C and RF set to 30%. Separation was achieved on a 50 cm × 75 μm column packed with C18AQ (1.9 μm; Dr Maisch, Ammerbuch, Germany) (PepSep, Marslev, Denmark) over 30 min at a flow rate of 300 nL/min. The peptides were eluted over a linear gradient of 3–23% Buffer B (Buffer A: 0.1% formic acid; Buffer B: 80% v/v acetonitrile, 0.1% v/v FA) and the column was maintained at 50 °C. The instrument was operated in data-dependent acquisition (DDA) mode with an MS1 spectrum acquired over the mass range 350–1500 m/z (120,000 resolution, 1 × 106 AGC and 50 ms maximum injection time) followed by MS/MS analysis with fixed cycle

time of 1.2 s via HCD fragmentation mode and detection in the orbitrap (30,000 resolution, $2 \times 10^5$ AGC, 54 ms maximum injection time, and 1.6 m/z isolation width). Only ions with charge state 2-6 triggered MS/MS with peptide monoisotopic precursor selection and dynamic exclusion enabled for 30 s at 10 ppm.

**Targeted phosphoproteomic analysis of AFF4 from Kusa 4B10 osteoblasts.** Phosphopeptides were analyzed on a Dionex 3500 nanoHPLC coupled to an Orbitrap Exploris 480 mass spectrometer (ThermoFischer Scientific) via electrospray ionization in positive mode with 1.9 kV at 275 °C and RF set to 30%. Separation was achieved on a 50 cm × 75 μm column packed with C18AQ (1.9 μm; Dr Maisch, Ammerbuch, Germany) (PepSep, Marslev, Denmark) over 60 min at a flow rate of 300 nL/min. The peptides were eluted over a linear gradient of 3–23% Buffer B (Buffer A: 0.1% formic acid; Buffer B: 80% v/v acetonitrile, 0.1% v/v FA) and the column was maintained at 50 °C. The instrument was operated in hybrid parallel reaction monitoring and data-independent acquisition (PRM/DIA) mode with an MS1 spectrum acquired over the mass range 350–1400 m/z (60,000 resolution, $1 \times 10^6$ automatic gain control (AGC) and 50 ms maximum injection time) followed by MS/MS analysis of $50 \times 13.7$ m/z isolation windows over the mass range of 360.5–1033.5 m/z via HCD fragmentation mode and detection in the orbitrap (30,000 resolution, $1 \times 10^6$ AGC, 55 ms maximum injection time; 30% normalized collision energy). Four additional scans were acquired per duty cycle targeting the mouse AFF4 S831 phosphopeptides (TVSpQSpSSLK, 548.719795 $m/z$, [M + 2H] 2 + ; TVSpQSSSLK, 508.736629 $m/z$, [M+2H]2 + ; RTVSpQSpSSLK, 626.77035 $m/z$, [M+2H]2 + ; RTVSpQSSSLK, 586.787185 $m/z$, [M + 2H] 2 + ), via HCD fragmentation mode and detection in the orbitrap (30,000 resolution, $1 \times 10^5$ AGC, 84 ms maximum injection time, and 2.0 m/z isolation width).

**Proteomic analysis of Kusa 4B10 osteoblasts.** Peptides were analyzed on a Dionex 3500 nanoHPLC coupled to an Orbitrap Eclipse mass spectrometer (ThermoFisher Scientific) via electrospray ionization in positive mode with 1.9 kV at 275 °C and RF set to 30%. Separation was achieved on a 50 cm × 75 μm column packed with C18AQ (1.9 μm; Dr Maisch, Ammerbuch, Germany) (PepSep, Marslev, Denmark) over 75 min at a flow rate of 300 nL/min. The peptides were eluted over a linear gradient of 3–23% Buffer B (Buffer A: 0.1% formic acid; Buffer B: 80% v/v acetonitrile, 0.1% v/v FA) and the column was maintained at 50 °C. The instrument was operated in data-independent acquisition (DIA) mode with an MS1 spectrum acquired over the mass range MS1 spectrum acquired over the mass range 350–950 m/z (60,000 resolution, $1 \times 10^6$ automatic gain control (AGC) and 50 ms maximum injection time) followed by MS/MS analysis of $38 \times 16$ $m/z$ isolation windows with a 1 $m/z$ overlap over the mass range of 349.5–950.5 m/z via HCD fragmentation mode and detection in the orbitrap (30,000 resolution, $1 \times 10^6$ AGC, 54 ms maximum injection time; 30% normalized collision energy).

**Immunoprecipitated AFF4 for interacting protein analysis.** Peptides were analyzed on a Dionex 3500 nanoHPLC coupled to an Orbitrap Lumos mass spectrometer (ThermoFisher Scientific) via electrospray ionization in positive mode with 1.9 kV at 275 °C and RF set to 30%. Separation was achieved on a 50 cm × 75 μm column packed with C18AQ (1.9 μm; Dr Maisch, Ammerbuch, Germany) (PepSep, Marslev, Denmark) over 90 min at a flow rate of 300 nL/min. The peptides were eluted over a linear gradient of 3–23% Buffer B (Buffer A: 0.1% formic acid; Buffer B: 80% v/v acetonitrile, 0.1% v/v FA) and the column was maintained at 50 °C. The instrument was operated in data-dependent acquisition (DDA) mode with an MS1 spectrum acquired over the mass range 300–1600 m/z (60,000 resolution, $1 \times 10^6$ automatic gain control (AGC) and 50 ms maximum injection time) followed by MS/MS analysis with fixed cycle time of 3 s via HCD fragmentation mode and detection in the orbitrap (15,000 resolution, $1.5 \times 10^5$ AGC, 54 ms maximum injection time, and 1.6 m/z isolation width). Only ions with charge state 2–6 triggered MS/MS with peptide monoisotopic precursor selection and dynamic exclusion enabled for 30 s at 10 ppm.

## LC-MS/MS data processing and analysis
**Mouse bone.** Phosphoproteomic data were searched against the UniProt mouse database (August 2020; UP000000589_109090 and UP000000589_109090_additional) with MaxQuant v1.6.7.0 using default parameters with peptide spectral matches, peptide and protein false discovery rate (FDR) set to 1%[89]. All data were searched with oxidation of methionine, and phosphorylation of Serine, Threonine and Tyrosine set as the variable modification, and carbamidomethylation of Cysteine and TMT to peptide N-termini and Lysine set as a fixed modification. First search MS1 mass tolerance was set to 20 ppm followed by recalibration and main search MS1 tolerance set to 4.5 ppm, while MS/MS mass tolerance was set to 20 ppm. Total proteomic data were searched against the UniProt mouse database (August 2020; UP000000589_109090) with SequestHT within Proteome Discoverer v2.5.0.4 (PMID: 24226387). The precursor MS tolerance was set to 20 ppm and the MS/MS tolerance was set to 0.02 Da with a maximum of 2 miss-cleavage. The peptides were searched with oxidation of methionine set as variable modification, and carbamidomethylation of cysteine and TMT to peptide N-termini and Lysine set as a fixed modification set as a fixed modification. Peptide spectral matches were filtered to 1% FDR using a target/decoy approach with Percolator (PMID: 17952086). The filtered PSMs from each database search were grouped and q-values generated at the peptide level with the Qvality algorithm. Finally, the grouped peptide data was further filtered to 1% protein FDR using Protein Validator. Quantification was performed with the reporter ion quantification node for TMT quantification in Proteome Discoverer. TMT precision was set to 20 ppm and corrected for isotopic impurities. Only spectra with <50% co-isolation interference were used for quantification with an average signal-to-noise filter of > 10. Data were processed with Perseus[90] to remove decoy data, potential contaminants and proteins only identified with a single peptide containing oxidized methionine. The Expand Site function was additionally used for phosphoproteomic data to account for multi-phosphorylated peptides prior to statistical analysis. Data were Log2-transformed followed by median-based normalization and differential expression performed with limma moderated t-test with Benjamini Hochberg FDR in NormalizerDE[91]. Known kinase-substrates relationships (KSRs) were retrieved from the PhosphoSitePlus database which includes orthologous sites mapped between mouse and human (Kinase_Substrate_Dataset; release date March 2022) followed by enrichment analysis in Perseus using Fisher's exact test without multiple testing hypothesis adjustment given the overall low number of associations identified. These known KSRs from PhosphoSitePlus were used as a training set for kinase substrate prediction using positive-unlabeled ensemble learning (KSP-PUEL)[30]. Position-specific scoring matrices were used to generate motif scores using +/− 6 amino acids flanking the phosphorylation site (total of 13 amino acids) of the training set using default settings. The motif score combined with the Log2(fold-change) across the various groups were used to train the model with ensemble size set to 50 and radial kernel type set in the KSP-PUEL GUI. GSEA of KEGG[92] and with ChEA3[49] was performed with TeaProt[93]. Protein:protein interaction analysis was performed with the STRING database[94].

**Immunoprecipitated AFF4 for phosphosite mapping, Akt/P7OS6K inhibitor analysis and P7OS6K in vitro kinase assay.** Data were searched against the UniProt mouse database (October 2021; UP000000589_109090) with SequestHT within Proteome Discoverer v2.5.0.4[95]. The precursor MS tolerance was set to 20 ppm and the MS/MS tolerance was set to 0.02 Da with a maximum of 2 miss-cleavage.

The peptides were searched with oxidation of methionine, and phosphorylation of Serine, Threonine and Tyrosine set as variable modification and carbamidomethylation of cysteine set as a fixed modification. Data were filtered to 1% FDR using a target/decoy approach with Percolator[96]. Phosphosite localization was performed with PhosphoRS[97]. Data were quantified with the Precursor Ion Quantifier node using default settings and statistical analysis performed in GraphPad Prism v10.0.

**Zebrafish caudal fin for phosphoproteomic analysis of P70S6K knockdown.** DDA data for spectral library generation was searched against the *Danio rerio* UniProt database (April 2021; UP000000437_7955 and UP000000437_7955_additional) with MaxQuant v1.6.12.0 using default parameters with peptide spectral matches, peptide and protein false discovery rate (FDR) set to 1%[89]. All data were searched with oxidation of methionine and phosphorylation of Serine, Threonine and Tyrosine set as the variable modification and carbamidomethylation set as a fixed modification. First search MS1 mass tolerance was set to 20 ppm followed by recalibration and main search MS1 tolerance set to 4.5 ppm, while MS/MS mass tolerance was set to 20 ppm. DIA data were processed with Spectronaut v16.0.220606 using the DDA spectral library with default parameters with precursor and protein Qvalue cutoff set to 0.01. PTM localization was enabled, probability cut-off set to 0.75 and summed PTM consolidation enabled. Peptide quantification was carried out at MS2 level using 3-6 fragment ions, with automatic interference fragment ion removal as previously described[98]. Dynamic MS1 and MS2 mass tolerance was enabled, and retention time calibration was accomplished using local (non-linear) regression. The default dynamic extracted ion chromatogram window size was performed. Data were processed with Perseus[90] to remove decoy data, potential contaminants and proteins only identified with a single peptide containing oxidized methionine. Transformation of the Spectronaut normal report was performed using the Perseus Plugin Peptide Collapse function to obtain phosphosite-level quantification[99]. Data were Log2-transformed followed by median-based normalization. Data were filtered to only contain phosphosites in the P70S6K motif (RxRxxpS/T). Differential expression analysis was performed with unpaired *t*-test with permutation-based FDR. For integration of mouse and zebrafish phosphoproteomic data, mouse phosphosite sequences, containing the phosphorylation site, surrounded by 15 amino acids were converted into a FASTA file format as the input queries for the protein BLAST analysis, performed using the NCBI blast+ software (version 2.10.1)[100]. The fasta sequences for all *Danio rerio* proteins were downloaded from the Uniprot database to produce a BLAST database (accessed on 27/05/2020). The blastp alignment was performed using a windowsize of 40, and word threshold and size of 11 and 2, respectively. For scoring, we used the PAM30 matrix, and gap opening and extension costs of 13 and 3, respectively. The alignment results were filtered to keep rows with *e*value < 0.01, a match between the Mouse and Zebrafish gene, and matching sequence of the phosphorylation site and surrounding amino acids.

**Zebrafish caudal fin for proteomic analysis of AFF4 knockdown.** Data was searched against the *Danio rerio* UniProt database (September 2025; UP000000437_7955 and UP000000437_7955_additional) and was processed with Spectronaut v16.0.220606 using directDIA with default parameters with precursor and protein Qvalue cutoff set to 0.01. Peptide quantification was carried out at MS2 level using 3-6 fragment ions, with automatic interference fragment ion removal as previously described[98]. Dynamic MS1 and MS2 mass tolerance was enabled, and retention time calibration was accomplished using local (non-linear) regression. The default dynamic extracted ion chromatogram window size was performed. Data were processed with Perseus v1.6.15[90] with Log2-transformation followed by median-based normalization. Differential expression analysis was performed with unpaired *t*-test with Benjamini−Hochberg FDR with significance cutoff set at *q* < 0.05. Pathway enrichment analysis was performed using FishEnrichr[101] focusing on GO Biological Processes ontology, where statistical significance was calculated with Fisher's exact test and the significance cutoff set at *p* < 0.05.

**Targeted phosphoproteomics of AFF4 S831 phosphorylation from Kusa 4B10 osteoblasts.** Data were processed using Skyline v21.1.0.146 and spectral libraries built with the Proteome Discovery.msf files from the AFF4 immunoprecipitation experiment described above using BiblioSpec[102,103]. Precursor and product ion extraction ion chromatograms (XICs) were generated using extraction windows two-fold the full width at half-maximum for both MS1 and MS2 filtering. Ion match tolerance was set to 0.055 m/z and matched to charges 2+ and 3+ for MS1 filtering of the first three isotopic peaks and 1+ and 2+ for MS2 filtering of b- and y-type ions. All data were manually confirmed for co-elution of MS1 and MS2 and have been uploaded to the Panorama Repository (see Deposited Data in Supplementary Table 1). Quantification was performed with nine MS2 fragments and only the singly non-miss-cleaved phosphopeptide was used for statistical analysis in GraphPad Prism v10.0 (TVSpQSSSLK, 508.736629 m/z, [M + 2H]2 + ).

**Proteomic analysis of Kusa 4B10 osteoblasts.** Data were processed with Spectronaut v16.0.220606 using directDIA with default parameters with precursor and protein Qvalue cutoff set to 0.01. Peptide quantification was carried out at MS2 level using 3−6 fragment ions, with automatic interference fragment ion removal as previously described[98]. Dynamic MS1 and MS2 mass tolerance was enabled, and retention time calibration was accomplished using local (non-linear) regression. The default dynamic extracted ion chromatogram window size was performed. Data were processed with Perseus[90] with Log2-transformation followed by median-based normalization. Differential expression analysis was performed with unpaired *t*-test with permutation-based FDR.

**Immunoprecipitated AFF4 for interacting protein analysis.** Data were searched against the UniProt mouse database (May 2022; UP000000589_109090 and UP000000589_109090_additional) with MaxQuant v1.6.7.0 using default parameters with peptide spectral matches, peptide and protein false discovery rate (FDR) set to 1%[89]. All data were searched with oxidation of methionine set as the variable modification and carbamidomethylation set as a fixed modification. First search MS1 mass tolerance was set to 20 ppm followed by recalibration and main search MS1 tolerance set to 4.5 ppm, while MS/MS mass tolerance was set to 20 ppm. Data were processed with Perseus[90] to remove decoy data, potential contaminants and proteins only identified with a single peptide containing oxidized methionine. A two-step statistical analysis was performed. First, data were Log2-transformed followed by median-based normalization to compare the enrichment of proteins in FLAG-AFF4-WT versus RFP negative control. Proteins quantified in <50% of the FLAG-AFF-WT replicates were discarded. Next, proteins quantified in 0/4 or 1/4 RFP negative control replicates were imputed using random values from the normal distribution with 0.5 width and 0.5 down shift and statistical analysis performed using unpaired *t*-test with permutation-based FDR. Only proteins significantly enriched in AFF4-WT versus RFP were included in downstream analysis. The second analysis normalized the level of AFF4-WT and AFF4-A-mutant to ensure that proteins differentially interacting were not due to subtle differences in the amount of enriched WT versus A-mutant. Statistical analysis was performed with an unpaired *t*-test with Benjamini−Hochberg-based FDR.

## Total RNAP2 ChIP-seq sample preparation, processing and analysis

Kusa 4B10 SSCs were differentiated into osteoblasts and made insulin-resistant as described above. The next day, cells were withdrawn from serum by incubating in MEMalpha, Pyruvate, GlutaMAX containing 0.2% BSA for 2 h to achieve a basal state. Cells were left unstimulated or stimulated with 10 nM insulin for 20 min. Cells were washed twice with cold PBS and then fixed with 1% paraformaldehyde for 10 min. Cross-linking was quenched with 125 mM glycine for 5 min and cells washed twice with cold PBS. Cells were scraped in Lysis Buffer 1 (50 mM HEPES-KOH pH 7.5, 140 mM NaCl, 1 mM EDTA, 10% glycerol, 0.5% NP40, 0.25% Triton X-100, containing 10 mM sodium flouride, 10 mM sodium pyrophosphate, 10 mM glycerophosphate, 2 mM sodium orthovanadate and EDTA-free PIC) and rotated at 4 °C for 5 min to break plasma membrane. Nuclei were pelleted at 2,000 × $g$, for 5 min at 4 °C and resuspended in Lysis Buffer 2 (50 mM HEPES-KOH pH 7.5, 140 mM NaCl, 1 mM EDTA, 1% Triton X-100, 0.1% SDS, 0.1% sodium deoxycholate containing 10 mM sodium flouride, 10 mM sodium pyrophosphate, 10 mM glycerophosphate, 2 mM sodium orthovanadate and EDTA-free PIC). Nuclei were lysed and DNA shearing performed by sonication in a Q-Sonica at 70% amplitude 15 s on / 45 s off for 20 min at 4 °C followed by centrifugation at 8000 × $g$ for 10 min at 4 °C. Protein was quantified with Pierce™ BCA (ThermoFisher Scientific) and 1.2 mg of enriched nuclear protein normalized to 300 μl of Lysis Buffer 2. An additional 30 μl of sonicated lysate was also saved as "input". Two micrograms of anti-RNAP2 antibody (SantaCruz; sc-56767) was added to the 1.2 mg normalized lysates and rotated overnight at 4 °C. The next day, 50 μl of protein-G beads (Invitrogen; #10004D) was washed with 1 ml of Low Salt Wash (20 mM Tris pH 8.0, 150 mM NaCl, 1 mM EDTA, 1% Triton X-100, 0.1% SDS) and the anti-RNAP2:protein lysate complexes were added to the beads and rotated for 1 h at 4 °C. The beads were washed with 1 ml of Low Salt Wash followed by 1 ml of High Salt Wash (20 mM Tris pH 8.0, 500 mM NaCl, 1 mM EDTA, 1% Triton X-100, 0.1% SDS) and finally 1 ml of LiCl Wash (20 mM Tris pH 8.0, 250 mM LiCl, 1 mM EDTA, 1% NP40, 0.1% SDS). Complexes were eluted with 120 μl of ChIP Elution Buffer (1% SDS, 100 mM NaHCO$_3$, pH 7.9) at 30 °C for 15 min while shaking at 900 x rpm. Ninety microliters of ChIP Elution Buffer was also added to the 30 μl aliquots of "input". Sodium chloride was added to the "input" and enriched samples by adding 4.8 μl of 5 M solution of NaCl to give a final concentration of 200 mM, and treated with 20 μg of RNAase A (Cell Signaling Technologies; #7013) for 30 min at 37 °C. The samples were further digested with 40 μg of Proteinase K (Cell Signaling Technologies; #10012) and de-crosslinked overnight at 65 °C. DNA fragments were purified using ChIP DNA Clean & Concentrator Kit (Zymo Research; D5205), quantified by Qubit dsDNA high sensitive kit (ThermoFisher Scientific; Q33230) and normalized to 4 ng/8 μl. Library preparation was completed using IDT xGen cfDNA and FFPE library preparation v2 (IDT rebranded the kit from Prism) as per manufacturer's instructions. Libraries were QC'd with an Agilent Tapestation with D1000 assay and quantitated using qPCR. Sequencing was completed with a NovaSeq 6000 using one lane of an S1-300 flow cell, XP workflow and v1.5 chemistry. Reads in FASTQ files were trimmed for adapters using Trimmomatic (v.39). Trimmed reads were aligned to the human reference genome (GRCh38) using Dragen OS tool provided by Illumina. Samtools (v1.16.1) was used to process bam files. Peak calling was performed using MACS2 (v2.2.8). Peaks near transcription start sites (TSS) were annotated using bedtools closest (v2.30.0). The bam-Coverage tool from deepTools (v3.5.1) was used to generate a normalized coverage track in BigWig format. Coverage was normalized using Reads Per Genomic Content (RPGC). Chromosome X was ignored for normalization to avoid bias from sex chromosomes. The enrichment pattern around TSS is visualized using plot heatmap utility provided by deepTools. The traveling ratio was calculated using the method described by ref. 15. Specifically for each gene, we calculated RNAP2 traveling index as:

$$RNAPII\ travelling\ ratio = \frac{Read\ count\ in\ TSS\ region/L1}{Read\ count\ in\ gene\ body/L2}$$

where the transcription start site (TSS) region of a gene is defined as the 50 bp to +300 bp around the TSS and the gene body is defined as +300 bp downstream of the TSS to +3 kb past the transcription end site. Differences in median RNAPII traveling indices were calculated using the Wilcoxon Rank Sum test where the number of asterisks denote the level of statistical significance: ***$p < 0.001$; **$p < 0.01$; and *$p < 0.05$.

## Quantitative PCR and analysis

Cells were lysed in trizol (#15596026; Thermo) by pipetting up and down and then centrifuged 12,000 × $g$ for 5 min at 4 °C. 1-bromo-3-chloroproane (B9673; Sigma) (1:100) was added to the supernatant and incubated for 5 min at room temperature and then centrifuged at 12,000 × $g$ for 15 min at 4 °C. The top phase is isolated and mixed with ethanol (1:1) and purified using the RNeasy mini kit columns (#74104; Qiagen) as per manufacturer instructions. RNA were normalized (1.2 μg) and cDNA synthesized using the high-capacity cDNA reverse transcription kit (#4374966; Thermo) and pre-amplified for Egr1, Fos, Myc and Gapdh (Hs00152928_m1, Hs04194186_s1, Hs00153408_m1 and Hs02786624_g1; Thermo) using the TaqMan preamp master mix (#391128, Thermo). Amplified cDNA were analyzed in technical triplicate on a Quantstudio 6 Flex (Thermo) using the TaqMan fast advance mix (#4444964; Thermo) for the same probes. mRNA levels were quantified using the delta-delta CT method and normalized to levels of GAPDH.

## Transcriptomic sample preparation, data processing and analysis

RNA was prepared as described above and total RNA quantity and quality were assessed using the high-sensitivity R6K ScreenTape system (Agilent# 5067-5582). An input of 500 ng total RNA for each sample was indexed separately using the TruSeq Prep RNA Kit v2 (Illumina) according to the manufacturer's instructions. Each library was quantified using the Agilent Tapestation (D1000 Screen Tape Agilent# 5067-5582). The indexed libraries were normalized to 1000 pm and subjected to cluster and paired-end sequencing (2 × 166 cycles) was performed using P2 300 cycle High Output Kit (Illumina) on a NextSeq 2000 Illumina instrument, yielding a minimum of ~40 million total reads per sample. Quality of raw sequencing data was evaluated using FASTQC (v 0.11.2) and mutiqc (v1.17). Adapters were trimmed using Trimmomatic (v0.39). Trimmed FASTQ files were aligned to the human reference genome (GRCh38) using HISAT2 (v2.2.1). Aligned reads were quantified using salmon (v1.10.0). GENCODE v45 annotations were used as a reference. Differential expression was performed with limma moderated t-test with Benjamini Hochberg FDR in NormalizerDE[91].

## Western-blot analysis

Protein was separated on NuPAGE 4–12% Bis-Tris protein gels (ThermoFisher Scientific) in MOPS SDS Running Buffer at 160 V for 1 h at room temperature. The protein was transferred to PVDF membranes (Millipore; #IPFL00010) in NuPAGE Transfer Buffer at 20 V for 1 h at room temperature and blocked with 5% skim milk in Tris-buffered saline containing 0.1% Tween-20 (TBST) for at least 30 min at room temperature with gentle shaking. The membranes were incubated overnight in primary antibody with 5% skim milk in TBST with gentle shaking at 4 °C and washed three times in TBST at room temperature. The membranes were incubated with HRP-secondary antibody in 5% skim milk in TBST for 45 min at room temperature and washed three times with TBST. Protein was visualized with Immobilon Western

Chemiluminescent HRP Substrate (Millipore; #WBKLS0500) and imaged on a ChemiDoc (BioRad). Densitometry was performed in ImageJ[104].

## Immunostaining and microscopy

For the nucleocytoplasmic distribution analysis, cells were transiently transfected with FLAG-tagged-AFF4-wild-type or A-mutant as described above, and treated with or without 100 nM insulin for 20 min, wash briefly in cold PBS and were fixed in 4% paraformaldehyde in PBS for 10 min at room temperature. The cells were then permeabilized for 10 min in PBS containing 0.5% Triton X-100, followed by incubation in a blocking solution (5% BSA in PBS) for 2 h. The cells were incubated with anti-FLAG antibody (F1804, Sigma) at a dilution of 1:1000 in blocking solution overnight at 4 °C. The following day, the cells were washed with PBS and incubated with Alexa-Fluor 488 conjugated secondary antibody (Invitrogen, A32723) at a dilution of 1:500 in blocking solution for 2 h. After secondary antibody incubation, the cells were washed with PBS and stained with DAPI before being mounted in Vectashield antifade aqueous mounting medium (Vector Laboratories). Images were acquired on a Zeiss LSM 880 confocal microscope equipped with an Airyscan module using a 40x oil (1.3 NA) objective lens and the emitted signal was detected using SuperResolution (SR) mode. Fluorescence images for each experiment were collected using identical settings. Deconvolved images were segmented using the IMARIS Cell module (version 9.6, Bitplane) to render a 3D version of the Z-stack and quantify volume-fill fluorescence intensity corresponding to the nucleus and cytoplasm. Fluorescence intensity and voxel data were exported using the Statistics tool in IMARIS.

## Zebrafish single guide RNA(sgRNA) design and synthesis

For each prioritized gene target, four sgRNAs were designed as a forward oligonucleotide that consisted of a T7 promoter recognition sequence, a target guide sequence, and an overlapping sequence complementary to a constant oligonucleotide (Sigma Aldrich, USA). Target guide sequences were either retrieved from a public database[105] or designed using CHOPCHOP v3.0 and synthesized by Sigma Aldrich, USA. Positive controls (*akt2, insr* and *entpd5*) and negative scramble (scr) control were also synthesized in parallel. Full sequence information for the forward oligonucleotides and constant oligonucleotide are provided in Supplementary Data 13. sgRNAs were generated in-house through in vitro transcription as previously described[105,106]. Briefly for each target gene, equimolar concentrations of the four (eight for duplicated genes) designed forward oligonucleotides and a constant oligonucleotide were annealed to generate a double-stranded DNA template comprising of the target sgRNA sequence. The DNA products were subsequently purified by the DNA Clean & Concentrator Kit (ZymoClean, USA) as per the manufacturer's instructions, eluted in 6 μL Rnase-free water, and their concentrations were determined by Nanodrop UV-Vis spectrophotometer (ThermoFisher Scientific, USA). All purified DNA templates had a final yield of 200–300 ng/μl. DNA templates were subjected to in vitro transcription using the T7 Ambion Megascript® kit (Invitrogen, USA) following the kit's instructions at 37 °C for 16 h, and any unused DNA template was digested using TURBO DNASe (Invitrogen, USA). The transcription products were purified by the RNA Clean & Concentrator Kit (ZymoClean, USA), eluted in 10 μL Rnase-free water and the final yield was determined using Nanodrop UV-Vis spectrophotometer (ThermoFisher Scientific, USA). All purified sgRNA had a final yield of 1000–2500 ng/μl and were diluted to 400-500 ng/μl working concentration prior to microinjection into zebrafish embryos.

## Zebrafish F0 crispant generation

**Microinjection.** 400–500 ng/μl sgRNA (candidate genes and negative scramble [scr]) was loaded onto 0.5 μg Alt-R® S.p. Cas9 nuclease protein (IDT, USA) by incubation at 37 °C for 5 min to form CRISPR/Cas9 ribonucleoprotein complexes. 0.05% phenol red was added to the injection mixture to aid in microinjection. Ribonucleoprotein complexes were loaded into pre-pulled glass capillary needles (Harvard Apparatus, USA) and the injections were performed on a PV820 Pneumatic Picopump (World Precision Instruments, USA) connected to a micromanipulator. The microinjection settings were as follows: the eject pressure was set at 30 psi, injection mode was set to timed mode, the solenoid open time was set as range = 100 ms and period between 1–2 ms. Bolus sizes were around 100 microns in diameter and uniform across both the negative scramble and the candidate gene injection to maintain experimental consistency. The ribonucleoprotein mixture was injected into the yolk sac of one-cell staged embryos. As our zebrafish are outbred, their growth depends on the genetics of the parent fish combined with environmental factors. For direct comparison, the candidate gene and negative scr injections were performed in parallel from the same embryo clutch, generated from the same breeding pair. Positive controls were treated exactly as the candidate genes. Injected embryos were monitored daily until 5 dpf and were transferred to an online flow system as described above to be raised as F0 crispants until 21 dpf.

**Genotyping.** Successful Cas9 gene edits were confirmed by genotyping the generated F0 crispants through polymerase chain reaction (PCR). For each gene (except *aff4*), four sets of primer pairs targeting each sgRNA recognition site were designed using Primer3Plus[107] and checked for off-target specificity using BLAT[108]. One primer pair was designed for *aff4* as all four sgRNA target the same exon. Full primer pair sequences have been detailed in Supplementary Data 13. Briefly, caudal fin clips of euthanized 21 dpf zebrafish crispants (n = 40–45 candidates, n = 5 scramble negative) were dissected and individually collected in a 96-well plate (Bio-Rad,USA). Genomic DNA was extracted, and target sequences were amplified by PCR using the MyTaq-Extract PCR Kit (Meridian Biosciences, UK). The generated amplicons were separated on a (w/v) 3% agarose gel in 1X TAE (40 mM Tris base, 20 mM acetic acid and 1 mM EDTA, pH 8.0) buffer at 120 V until dye front reached 75% of the gel length. The gel was stained with 0.05% RedSafe dye (Intron Biotechnology, Republic of Korea) to visualize the DNA bands and a 50 bp Hyperladder molecular marker (Meridian Biosciences, UK) was separated in parallel to assess amplicon size. The gel was imaged using the GelRed setting on ChemiDoc MP imaging system (Bio-Rad, USA). All Genotyping of the positive hits are shown in Supplementary Fig. 4.

***aff4* germline zebrafish line.** F0 *aff4* crispants were outcrossed to wildtype AB zebrafish to generate stable heterozygous F1 *aff4*[+/-] mutants with a 100 bp deletion in one allele that was confirmed with Sanger sequencing (Supplementary Fig. 5). = The generated F1 *aff4*[+/-] embryos were further raised to adulthood as described above. The associated F0 *aff4* parent fish was designated as a founder and maintained separately in our online flow system in 1 L tanks (Tecniplast, Italy) along with a buddy fish to minimize animal stress. At maturity, the F1 *aff4*[+/-] were setup as in-cross mating pairs to produce a F2 generation within Mendelian ratios of 23% wild-type, 48% heterozygous and 28% homozygous (*aff4*[-/-]) mutants confirmed by genotyping through PCR as described above and statistical significance was calculated using Chi-square ($X^2$) test. The produced F2 embryos were raised either to 21 dpf for bone staining or to adulthood as described above.

## Zebrafish cartilage and mineralization bone staining

Euthanized zebrafish F0 crispants were treated with alcian blue and/or alizarin red to mineralize the cartilage and bone, respectively. The staining was done as described previously (cartilage[109]; bone[110]), and performed in multiwell plates, where each plate consisted of F0 crispants from both treatment groups (candidate gene and associated

scramble negative) to avoid batch effects. All steps were performed on a plate shaker (Ratek, Australia) at 200 rpm unless stated otherwise. Briefly for the cartilage stain, 7 dpf zebrafish larvae were fixed in v/v 4% paraformaldehyde solution and stained overnight in w/v 0.02% alcian blue in 70% EtOH/80 mM $MgCl_2$ solution at room temperature (RT). The stained larvae were washed in saturated sodium tetraborate solution and residual tissue was digested in 0.5 µg/µl trypsin solution in w/v 60% saturated sodium tetraborate solution at RT for 2 h. Larvae were destained in v/v 20% tween-20/1% KOH solution at RT for 2 h and subsequently imaged. For the bone staining, euthanized and gut cleared 21 dpf zebrafish were fixed in v/v 5% formalin/ 5% Triton-X 100/ 1% KOH solution at 42 °C for 24 h, followed by treatment in v/v 20% ethylene glycol/1% KOH solution at 42 °C for 48 h to enhance skin transparency. The specimens were stained in w/v 0.05% alizarin red in 20% ethylene glycol/1% KOH solution for 30 min at RT, destained in v/v 20% tween-20/1% KOH solution at 42 °C for 3 h and consequently imaged. Prior to imaging, the destain solution was removed and all stained fish were dorsally oriented.

### Zebrafish image analysis and statistics

Brightfield imaging was performed on a M165FC Leica stereomicroscope, equipped with a DMC 4500 camera, and processed via the Leica Application Suite X, version 4.13.0 (Leica, Germany). The acquired raw images were exported as .tiff files and further analyzed on ImageJ version 2.3.0/153f51 to measure three parameters: fish length, bone mineralization and skeleton length. The raw files were individually checked for image integrity and images with interfering background noise were excluded from analysis. Zebrafish body length and bone mineralization were analyzed using a semi-automated ImageJ plugin ZFBONE[111]. Body length was defined as standard length and measured from the mouth tip to the caudal vertebra base using the STU feature of ZFBONE. Additionally, the skeleton length of each fish was measured using the ImageJ skeleton length plugin[112]. The exported data for each parameter were normalized to the negative scramble control and statistical analyses were performed on GraphPad Prism v 10.2.3. Unpaired Student's $t$-test was performed between the candidate gene and negative scramble F0 crispants to calculate standard error of the mean and $p$ value significance with the significance cut-off set at $p < 0.05$. Full statistics for the dataset are provided in Supplementary Data 4. For the alcian blue stained fish, the .tiff raw image files were processed using ImageJ and only the head region was used for analysis. The images were converted to 8-bit binary format, pixel thresholding was performed to minimize background noise and the head boundary was defined as the region of interest (ROI). The alcian blue mean pixel intensity and the ROI area were measured and exported for data normalization and univariate statistical analysis as described above. The results were plotted as a ratio of the mean pixel intensity to the ROI area to analyze differences in cartilage development. Data visualization for all data sets was done using GraphPad Prism v 10.2.3 and Adobe Illustrator v 26.0.1. For the stable *aff4* mutant line ZFBONE dataset, One-way ANOVA with Benjamini-Hochberg correction was performed between all 3 treatment groups and q-value significance was calculated with significance cut-off set at $q < 0.05$. Data was plotted as violin plots using GraphPad Prism v 10.2.3 and Adobe Illustrator v 26.0.1.

### Resource availability

Further information and requests for reagent and resources may be directed to and will be fulfilled by the Lead Contact, A/Prof Benjamin L. Parker (ben.parker@unimelb.edu.au).

### Reporting summary

Further information on research design is available in the Nature Portfolio Reporting Summary linked to this article.

## Data availability

The phosphoproteomics and proteomics data generated in this study are deposited to the ProteomeXchange Consortium (http://proteomecentral.proteomexchange.org/cgi/GetDataset) via the PRIDE[113] and can be accessed through the links provided below: PRIDE: PXD054205 (Phosphoproteomic and proteomics of insulin signaling in aged mouse bone). PXD054212 (Zebrafish caudal fin phosphoproteomic of Rps6kb1a/b knockdown). PXD054247 (Analysis of insulin-regulated phosphorylation of AFF4 with Akti or S6Ki and S6K in vitro kinase assay). PXD054250 Affinity purification – mass spectrometry of AFF4 WT or S829/S831/3/5/8 A mutant). PXD054479 (Proteomic and secretomic analysis of Kusa 4B10 osteoblasts). The targeted proteomic data can be accessed via Panorama Web Repository[114] through the accession code Panorama Web: U of Melbourne – Parker Lab: PRM of mouse AFF4 S831 phosphorylation (Targeted phosphoproteomics of mouse S831 AFF4 phosphorylation in control or insulin resistant osteoblasts). The transcriptomic dataset generated in this study is deposited to NCBI and can be accessed through the accession code below: PRJNA1146056 (Transcriptomics of HEK293T cells expressing AFF4-wild type or AFF4-T829/S831/S833/S834/S835A mutant treated with or without insulin). Full list of Deposited Data is also provided in Supplementary Table 1. Source data are provided with this paper.

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

## Acknowledgements

We thank Nicholas Williamson, Ching-Seng Ang, Shuai Nie, Swati Varshney and Michael Leeming for instrument support in the Bio21 Mass Spectrometry and Proteomics Facility. We thank Mr. Cameron Mackey and Mr. Bryan Ko for providing technical support for zebrafish maintenance. This research was supported by access to the Melbourne Mouse Metabolic Phenotyping Platform at The University of Melbourne. This work was funded by a University of Melbourne Driving Research Momentum Grant, an NHMRC Emerging Leader Investigator Grant APP2009642 and Australian Research Council Discovery Grant DP250100201 to B.L.P. G.T.D is funded by NHMRC Grants 2022/GNT2021126, 2020/GNT2002427, 2018/GNT1160043, Australian Research Council Grant DP220102910, Diabetes Australia Grants Y23G-DodG, Y20G-DodG and The University of Melbourne Deans Innovation Award. S.J.V. is supported by a CSL Centenary Fellowship and SNOW Medical Fellowship. We thank Prof Benjamin M. Hogan and Mr. Scott Paterson from the Peter MacCallum Cancer Center, Australia for assistance with the zebrafish functional screen. We thank Dr Marco Tarasco from The University of Algarve, Portugal, for designing the ZFBONE brightfield ImageJ macro used in the zebrafish bone mineralization analysis.

## Author contributions

B.L.P. conceptualized the study. M.D., L.L., R.B., A.C., A.B.-G., L.D., I.S.R., N.K.Y.W., V.R.H., V.U., J.P.H.W., B.L.P. performed the experiments. M.D., L.L., H.J.K., R.B., A.C., H.K., A.B.-G., J.M., P.Y., B.L.P. analyzed the data. G.S.L., K.A.S., M.K.M., M.J.W., G.T.D., S.J.V., N.A.S., B.L.P provided resources, guided experimental design, supervised and funded the research.

## Competing interests

The authors declare no competing interests.

## Additional information

[1]Department of Anatomy and Physiology, School of Biomedical Sciences, Faculty of Medicine, Dentistry and Health Sciences, The University of Melbourne, Parkville, VIC, Australia. [2]Centre for Muscle Research, Department of Anatomy and Physiology, The University of Melbourne, Victoria, VIC, Australia. [3]Computational Systems Biology Group, Children's Medical Research Institute, Faculty of Medicine and Health, The University of Sydney, Westmead, NSW, Australia. [4]Charles Perkins Centre, School of Mathematics and Statistics, The University of Sydney, Sydney, NSW, Australia. [5]The Kinghorn Cancer Centre and Cancer Research Theme, Garvan Institute of Medical Research, Darlinghurst, NSW, Australia. [6]School of Clinical Medicine, Faculty of Medicine and Health, University of New South Wales, Sydney, NSW, Australia. [7]The Walter and Eliza Hall Institute of Medical Research, Parkville, VIC, Australia. [8]Department of Medical Biology, The University of Melbourne, Parkville, VIC, Australia. [9]St. Vincent's Institute of Medical Research, and Department of Medicine at St. Vincent's Hospital, The University of Melbourne, Fitzroy, VIC, Australia. ✉e-mail: ben.parker@unimelb.edu.au

