## [Transparent Peer Review file · Nature Communications]

Phosphoproteomics of aged insulin-resistant bone identifies P70S6K phosphorylation of AFF4 as a gene-specific transcriptional regulator

Corresponding Author: Dr Benjamin Parker

Version 0:

Reviewer comments:

Reviewer #1

(Remarks to the Author)

General Comments

The authors conducted a phosphoproteomic screen of bone tissue from young, lean, insulin-sensitive mice versus old, obese, insulin-resistant (IR) mice, all acutely treated with insulin. Differentially regulated proteins were then functionally investigated using a zebrafish genomic screen focused on bone development phenotypes. Among the findings, AFF4—a component of the super elongation complex (SEC)—was identified as a substrate of P70S6K. A specific phosphosite on AFF4, serine 831 (S831), was found to be dysregulated in cultured osteoblasts presumed to be insulin resistant following high-dose insulin treatment. Functional analyses suggest that phosphorylation at S831 is required for the recruitment of ENL/AF9 to histones and for initiating transcription. This study encompasses a large and complex dataset, offering important and novel insights into the molecular mechanisms through which insulin and its downstream signaling pathways may regulate gene transcription.

Despite the strengths of the study, including the integration of phosphoproteomics with functional genomic analysis, several issues—both technical and interpretational—need to be addressed for clarity and completeness.

Major Points

- Table 1 is missing and should be included to support the referenced data.
- Figure labeling and clarity: Several figures lack adequate labeling or explanation:
 - o Figure 1G (heatmap): Consider including the legend in the supplementary materials.
 - o Figures 4F and 6: Missing or unclear axis labels.
 - o Figures 2F–H: Insufficient detail about what is plotted; further clarification is needed.
- Insulin signaling in osteoblasts: Kusa 4B10 osteoblasts were treated with high insulin levels to induce insulin resistance. However, canonical insulin signaling markers should be shown in Western blots to confirm impairment of insulin signaling.
- Data accessibility: The PRIDE repository link does not work and should be corrected to allow verification and reuse of the dataset.

Specific Comments and Suggestions

- Kinase substrate coverage ("2.78%"): It is unclear how this value was derived. Please clarify:
 - o Which version of PhosphoSitePlus was used?
 - o Were only mouse-specific substrates included?
- Kinase prediction tool (KSP-PUDEL): This tool is relatively outdated (~10 years old). The findings should be validated using more recent prediction tools (e.g., PhosphoSitePlus's updated kinase-substrate tools), which reflect current phosphoproteomic knowledge.
- Figure 1I: Only known phosphosites from the training set are shown. It would strengthen the analysis to indicate where predicted sites—especially AFF4-S831—are located on the prediction/motif score scatterplot. To improve readability, consider removing some training set phosphosite labels.
- Figure 3A: The Venn diagram reports 25 genes, but the primary analysis described 22 genes. Please clarify whether the additional 3 genes were later included as controls.
- Bone-enriched proteome: Specify whether the inclusion criteria for the bone-enriched proteome are consistent with those used in the Human Protein Atlas or Mouse Proteome Atlas.

- Comparative proteomics: It would be helpful for non-specialists to see how similar the proteomes or phosphoproteomes are between mouse bone, Kusa 4B10 cells, and zebrafish bone tissue.
- Mouse fasting protocol: Information regarding the fasting duration prior to insulin injection is missing and should be reported.
- Insulin dosage: The insulin dose used in mice appears relatively high. Please justify this choice and, if available, include data on blood glucose responses (e.g., insulin tolerance test results).
- Insulin and mRNA processing: The discussion states it is "unknown whether insulin regulates other steps in mRNA production." However, recent studies have shown insulin impacts several mRNA processing steps, including splicing. These references should be cited.
- P70S6K knockout (KO) mice: Given the known literature on bone-related phenotypes in P70S6K KO mice, relevant references should be included to place findings in context.
- Figure 4J: The knockout of P70S6K has a greater effect on RPS6 than on AFF4. Please discuss whether other kinases may be involved in AFF4 regulation.
- RNAP2 traveling ratio (Figures 5H and 5J): There appears to be a discrepancy between the ChIP-seq track area under the curve (AUC) and the bar graph representation of RNAP2 distribution. Clarify how the traveling ratio was calculated and why RNAP2 appears primarily located in the promoter region.
- AFF4 phosphomutants: Multiple serine/threonine residues were mutated to alanine in the mutant AFF4 construct. Are there phosphopeptides in the dataset that support phosphorylation at these sites (e.g., T829, S833, S834, S835)?
- AFF4 downstream targets (Figure 6): Several downstream targets of AFF4 were identified in HEK293T cells (e.g., Figures 5D–H). Please discuss whether these targets are cell-type specific and what their relevance might be for osteoblast function.

Reviewer #2

(Remarks to the Author)

This manuscript describes the insulin-stimulated phosphoproteome of bone from old vs young mice. The authors validated several candidates in zebrafish and showed that AFF4 knockout severely affected bone growth in zebrafish. The authors performed a robust validation to show that insulin-stimulated phosphorylation of AFF4-S831 is involved in releasing paused RNAP2 to initiate elongation of select transcripts. This study significantly advances our knowledge of the role of insulin action on AFF4 and release of RNAP2 pausing.

While the validation part of the study to show the role of AFF4 in elongation of select transcripts is very thorough, there are several gaps, such as the criteria for selection of candidates as well as selection of AFF4 for validation. Another gap is the missing link of AFF4 phosphorylation to bone function.

Issues:

The flow of the manuscript is somewhat interrupted by describing phosphoproteomics first, then describing the total proteome, the bone-enriched proteome, the secretome, the enriched transcription factors (but only downregulated proteins), before reverting to the phosphoproteome. No rationale is provided for these diversions, and the only part that may be relevant for the AFF4 part is the enriched transcription factor analysis, but this is not referred to when potentially relevant (after Figure 6). Reordering or even deleting some less relevant parts may help improve the flow. For example, the relevance of the secretome analysis to the rest of the manuscript is not clear unless you can somehow link this to bone function and/or the phosphoproteomic data. E.g. are the proteins mentioned in lines #244-247 bone enriched proteins? And how they relate to the DE phosphoproteins? E.g. Is BMP signalling and WNT signalling also changed in phospho?

While normalising the phosphoproteome to total proteome in principle makes sense, discarding data from 1/3rd of all proteins may omit crucial findings, particularly when focusing on insulin-stimulation. Also the numbers of differentially regulated phosphosites are very confusing in lines #147-150, with 650 or 1004 differentially regulated sites? It is also not clear whether the candidate proteins were identified in the total proteome.

It is surprising that the criteria for candidate prioritisation did not include a change in insulin responsive phosphorylation between young and old, when this was the major aim of the study?

The criteria for positive and negative hits in the zebrafish screen are not clear as some of the negatives appeared to have a stronger phenotype than AFF4? In Figure 3B it is difficult to determine which parameters are significant. The selection of AFF4 above any of the other positive hits would benefit from a rationale, particularly since the growth phenotype in Figure 3G was not that marked.

The calculation of the travelling ratio seems odd as it is counterintuitive that a low travelling ratio reflects less paused RNAP2 (lines 406-410). The data in figure 5F suggests that ~50% of all RNAP2 transcripts are affected in IR and looks very convincing, but the data presented in 5G-J is much less convincing. Particularly Fig 5G (and also 6C) is difficult to interpret as so many dots (many of which are probably significant) are overlaid and clustered close to zero and maybe alternate visualisation possibilities should be explored.

Is there any overlap of affected genes between the experiments in Fig 5E-G and Fig 6C?

Why were Egr1 and Fos selected in Figure 6O?

Confirmation that the AFF4 – H3K18Cr was prevented with the inhibitor would strengthen the confidence into the inhibitor. Can the role of ELL and ELL2 be excluded as there is a trend for these (significant in Fig 6K but only n=2 data points in Fig 6L)?

Is there any evidence for a link between histone crotonylation and any AFF4-A affected transcripts?

The model relies on AFF4 phosphorylation to occur prior to recruitment to crotonylated histones via ENL/AF9 but how does this agree with the pausing of the already DNA-bound RNAP2?

How does the AFF4 mediated changes in transcription lead to the bone defects observed in Fig 3G and 3I? E.g. does AFF4 regulate any bone enriched proteins and/or any bone-specific secreted proteins (Fig 2A, 2D)?

Minor issues:

The rib figure was a bit confusing as 18+ maybe is meant to be <18?

Please explain why the scr fish differ so vastly in length (3G vs 4G).

Version 1:

Reviewer comments:

Reviewer #1

(Remarks to the Author)

The revised manuscript has been edited for improved clarity. As requested, new experimental data have been included, and the authors have addressed the issues raised by this reviewer. The authors have strengthened their case, and the manuscript has improved considerably. I have no further comments.

Reviewer #2

(Remarks to the Author)

The authors have addressed all issues raised by this reviewer. They have improved the manuscript including the flow, and clarified several sections including the selection criteria. The authors could consider adding a sentence or two at line 563 in the discussion on the crotonylation status of the proteins that were downregulated in the AFF4 knockdown zebrafish, if such information is available, as an additional link of AFF4 phosphorylation to bone function. But this is just a suggestion and the manuscript is good as it is.

Minor issues:

Track changes on line 199.

Typo in line 604 (dependent)

Reviewer #3

(Remarks to the Author)

The manuscript by Dutt and Colleagues describes the role of AFF4, a component of the super elongation complex (SEC), in mediating insulin-regulated gene expression. By comparing the bone phosphoproteomes of young and aged insulin-resistant (IR) mice, the authors identified a list of candidate proteins whose phosphorylation levels are significantly different between the two groups. To assess the role of these candidates in bone biology, they disrupted the orthologous genes in zebrafish and identified several genes, including *aff4*, whose knockdown leads to bone defects. The author went on to demonstrate that AFF4 S831 phosphorylation is blunted in an IR cell model. They determine that P70S6K mediates the insulin-dependent phosphorylation of AFF4 S831 in HEK293T cells, and this phosphorylation is necessary for promoting the expression of insulin-regulated genes HEK293T and Kusa 4B10 cells. They further show that AFF4-pS831 recruits ENL/AF9 complex to the SEC to enhance transcription. Therefore, the study identified a new mechanism of insulin-regulated gene expression via modulating transcriptional elongation and demonstrated its deficiency in IR. The methodologies are sound, the results are of good quality, the conclusions are well supported, and the manuscript is clearly written. Nevertheless, there are a few issues.

Major issues:

- In the zebrafish studies, the mutagenesis rate of each sgRNA is not characterized. Although efficient mutagenesis of some sgRNAs is evident in Fig S4, the efficiency of most sgRNA cannot be discerned from the images. A TIDE analysis may be more informative. Such results would provide the readers with more confidence that the negative hits are true negatives. In the same vein, it would be great if the authors can provide the exact mutation in the germline *aff4* line they further characterized.

Minor issues:

- It is unclear why the *aff4* is prioritized in the zebrafish study. The KD phenotype of the *aff4* crispants is the weakest among the positive hits. In fact, it seems to be weaker than some of the negative hits.

- In the experiments presented in Figure 6, AFF4 mutant cells would be a better model for the overexpression of the WT and mutant AFF4.

Reviewer #1 (Remarks to the Author):

General Comments

The authors conducted a phosphoproteomic screen of bone tissue from young, lean, insulin-sensitive mice versus old, obese, insulin-resistant (IR) mice, all acutely treated with insulin. Differentially regulated proteins were then functionally investigated using a zebrafish genomic screen focused on bone development phenotypes. Among the findings, AFF4—a component of the super elongation complex (SEC)—was identified as a substrate of P70S6K. A specific phosphosite on AFF4, serine 831 (S831), was found to be dysregulated in cultured osteoblasts presumed to be insulin resistant following high-dose insulin treatment. Functional analyses suggest that phosphorylation at S831 is required for the recruitment of ENL/AF9 to histones and for initiating transcription. This study encompasses a large and complex dataset, offering important and novel insights into the molecular mechanisms through which insulin and its downstream signaling pathways may regulate gene transcription.

Despite the strengths of the study, including the integration of phosphoproteomics with functional genomic analysis, several issues—both technical and interpretational—need to be addressed for clarity and completeness.

We thank the reviewer for their positive comments and comprehensive review. All the suggestions are excellent which we believe have improved the manuscript.

Major Points

- *Table 1 is missing and should be included to support the referenced data.*

This was a typo and has been corrected to Table S1.

- *Figure labeling and clarity: Several figures lack adequate labeling or explanation:
o Figure 1G (heatmap): Consider including the legend in the supplementary materials.*

We have updated the legend for Figure 1:

“(G) Heatmaps showing the quantification of insulin-regulated phosphorylation in 10- and 73-week old bone that are known Akt/P70S6K and mTOR substrates, and used as the training set for machine learning predictions.”

- o *Figures 4F and 6: Missing or unclear axis labels.*

We apologise for this and have updated all axis labels.

- o *Figures 2F–H: Insufficient detail about what is plotted; further clarification is needed.*

We have updated the legend for Figure 2:

“(F) GSEA of the proteomics data using the transcription factor:target geneset annotations in the ChEA3 database. Data presents the normalized enrichment score (NES) focusing on proteins down-regulated in the 73-week old bone to estimate potential decreased transcription factor activity. Individual GSEA plots of the target

genes for FOS (G) and JUND (H) showing the target genes are significantly down-regulated where proteins are ranked from up-regulated to down-regulated.”

• *Insulin signaling in osteoblasts: Kusa 4B10 osteoblasts were treated with high insulin levels to induce insulin resistance. However, canonical insulin signaling markers should be shown in Western blots to confirm impairment of insulin signaling.*

This is a good point. We performed additional western-blot analysis of phospho-Akt and now show reduced insulin-dependent activation under insulin resistant conditions. These data are included in new Figures 5B-C and uncropped western blots provided in source Data S1. The results have been updated on line 407:

“IR attenuated insulin-induced phosphorylation of S473 on Akt along with pT389 on P70S6K and the substrate S235/6 on RPS6 (Figure 5B-C).”

• *Data accessibility: The PRIDE repository link does not work and should be corrected to allow verification and reuse of the dataset.*

The public link will become active upon acceptance of the manuscript. Reviewer access has been granted via a Reviewer Username and Password with all login details provided in the Key Resources Table. Please visit <https://www.ebi.ac.uk/pride/login> and access via reviewer log-in and password.

Note, there is separate login for each dataset e.g below screenshots for “Phosphoproteomic and proteomics of insulin signaling in aged mouse bone”:

PRIDE
Proteomics IDEntifications Database

Home Resources Tools Help License About Contact

Log In

① Login with your registered email address OR reviewer username (ex: reviewerXXX@ebi.ac.uk)

reviewer_pxd054205@ebi.ac.uk

8ZPOAq6TmF2

Show Password Forgot Password

Log In

PRIDE
Proteomics IDEntifications Database

Home Resources Tips Help License About Contact

reviewer_pxd054205@ebi.ac.uk Register

Private Project PXD054205

Transfer Ownership Edit Publish

Summary

Title
Phosphoproteomics and proteomics of insulin signaling in aged mouse bone

Description
We compared the in vivo insulin signaling response in bone tissue of 10- versus 73-week old C57BL/6J male mice (Figure 1A). Saline or insulin (2.5 mU/kg) was interperitoneally injected and mice sacrificed after 20 min. Tibia bone was rapidly dissected and flushed with ice-cold PBS within 30 sec to remove marrow and quench phosphorylation before being snap frozen in liquid nitrogen. Protein was ext...

Read more

Sample Processing Protocol
Tibia bones cleared of bone marrow cells were ground into powder under liquid nitrogen lysed 6 M guanidine HCl (Sigma, #G4505), 100 mM Tris pH 8.5 containing 10 mM Tris(2-carboxyethyl)phosphine (TCEP) (Sigma, #75259) and 40 mM 2-chloroacetamide (CAA) (Sigma, #Z2790) by tp-probe sonication. The lysate was heated at 95°C for 5 min and centrifuged at 20,000 x g for 10 min at 4°C. The supernatant wa...

Read more

Data Processing Protocol
Phosphopeptides and non-enriched peptides were analyzed on a Dionex 3500 nanoHPLC coupled to an Orbitrap Eclipse mass spectrometer (ThermoFischer Scientific) via electrospray ionization in positive mode with 1.9 kV at 275°C and RF set to 30%. Separation was achieved on a 50 cm x 75 µm column packed with C18AQ (1.9 µm; Dr Masch, Ammertbuch, Germany) (PepSep, Marslev, Denmark) over 60 min (fractiona...

Read more

Contact
Dr Benjamin Parker, The University of Melbourne
Dr Benjamin Parker, The University of Melbourne (lab head)

Submission Date
25/07/2024

Properties

Organism
Mus musculus (Mouse)

Organism part
tibia

Diseases
Unknown

Modification
phosphorylated residue

Instrument
Orbitrap Eclipse

Software
Unknown

Experiment Type
Bottom-up proteomics

Quantification
TMT

Specific Comments and Suggestions

- Kinase substrate coverage ("2.78%"): It is unclear how this value was derived. Please clarify:

- o Which version of PhosphoSitePlus was used?

- o Were only mouse-specific substrates included?

We have updated the methods on line 1059 to answer these questions:

“Known kinase-substrates relationships (KSRs) were retrieved from the PhosphoSitePlus database which includes orthologous sites mapped between mouse and human (Kinase_Substrate_Dataset; release date March 2022) followed by enrichment analysis...”

- Kinase prediction tool (KSP-PUEL): This tool is relatively outdated (~10 years old). The findings should be validated using more recent prediction tools (e.g., PhosphoSitePlus’s updated kinase-substrate tools), which reflect current phosphoproteomic knowledge.

PhosphoSitePlus does not have a prediction tool to estimate novel KSRs; it is a database of experimentally validated upstream kinases.

While the KSP-PUEL tool was published by co-author Pengyi Yang in 2016, the machine learning model was re-trained using a curated positive control training set of phosphosites identified in our data with known kinase-substrate relationships (KSRs) retrieved from the PhosphoSitePlus database (downloaded March 2022). We have modified the methods on line 1063 to clarify:

“These known KSRs from PhosphoSitePlus were used as a training set for kinase substrate prediction using positive-unlabeled ensemble learning (KSP-PUEL).

Position-specific scoring matrices were used to generate motif scores using +/- 6 amino acids flanking the phosphorylation site (total of 13 amino acids) of the training set using default settings. The motif score combined with the Log2(fold-change) across the various groups were used to train the model with ensemble size set to 50 and radial kernel type set in the KSP-PUEL”

• *Figure 1I: Only known phosphosites from the training set are shown. It would strengthen the analysis to indicate where predicted sites—especially AFF4-S831—are located on the prediction/motif score scatterplot. To improve readability, consider removing some training set phosphosite labels.*

Thank you for this suggestion. We have removed the labels of the training sets and listed some example top predicted substrates for Akt/P70S6K or mTOR. We have also added a citation to Table S1 to direct the readers to these data and the training set of known substrates. AFF4-S831 had one of the highest Akt/P70S6K predictions scores (0.99).

• *Figure 3A: The Venn diagram reports 25 genes, but the primary analysis described 22 genes. Please clarify whether the additional 3 genes were later included as controls.*

The Venn diagram shows the overlap of proteins that contain an orthologous phosphosite in zebrafish, those that have an Akt/P70S6K prediction >0.75, and those whose encoding gene are associated with BMD based on human GWAS. The final screen consisted of a subset of these that displayed differential insulin response in the young vs old mice. Of our 3 controls genes, only AKT2 has a conserved orthologous phosphosite in zebrafish that is insulin regulated while no phosphosites on ENTPD5 were identified.

• *Bone-enriched proteome: Specify whether the inclusion criteria for the bone-enriched proteome are consistent with those used in the Human Protein Atlas or Mouse Proteome Atlas.*

Yes, these criteria were based on the Human Protein Atlas. We have further clarified this on line 217:

“We used criteria established in the Human Protein Atlas based on >4-fold higher levels in bone compared to all other tissues to identify 292 bone enriched proteins (Figure 2A).”

• *Comparative proteomics: It would be helpful for non-specialists to see how similar the proteomes or phosphoproteomes are between mouse bone, Kusa 4B10 cells, and zebrafish bone tissue.*

While there is some value in comparing these model systems, we only have phosphoproteomic data on mouse bone and zebrafish tails where we performed an orthologous phosphosite analysis. Given that Reviewer 2 has questioned the overall flow and inclusion of mouse bone proteomics, we feel that additional comparative analysis would further disrupt the flow and extend the length of the manuscript.

• *Mouse fasting protocol: Information regarding the fasting duration prior to insulin injection is missing and should be reported.*

Mice were fasted for 12h and we have updated line 124:

“Mice were fasted for 12 h and saline or insulin (2.5 mU/kg) was interperitoneally injected, and mice were sacrificed after 20 min.”

• *Insulin dosage: The insulin dose used in mice appears relatively high. Please justify this choice and, if available, include data on blood glucose responses (e.g., insulin tolerance test results).*

While the insulin dose is relatively high, delivery was achieved via IP injection and levels in the circulation will be considerably lower. The same supraphysiological dose of 2.5 mU/kg has been used by several groups during IV delivery in hyperinsulinemic-euglycemic clamps (e.g PMID: 27238637, 20332342, 20393148, 11485987). We do not have insulin tolerance test data since mice were sacrificed after 20 min and to preserve phosphorylation levels in bone, we had to focus on extremely rapid dissection protocols (<1min). We can confirm that no mice were unconscious at the 20 min time-point indicating they did not experience pathological hypoglycemia.

• *Insulin and mRNA processing: The discussion states it is "unknown whether insulin regulates other steps in mRNA production." However, recent studies have shown insulin impacts several mRNA processing steps, including splicing. These references should be cited.*

Indeed, insulin has been shown to influence post-transcriptional regulation. We have modified the discussion on line 533 and provide reference to a recent excellent review:

“Insulin-dependent phosphorylation of transcription factors can activate the transcriptional cycle, and insulin has also been shown to regulate the post-transcriptional processing of mRNA^{70,71}. However, it is currently unknown whether insulin regulates other steps in mRNA production such as RNAP2 pause-release and transcriptional elongation. This is an important question because these later steps of the transcriptional cycle are emerging as the key regulatory checkpoints and rate-limiting steps of gene expression¹⁴.”

• *P70S6K knockout (KO) mice: Given the known literature on bone-related phenotypes in P70S6K KO mice, relevant references should be included to place findings in context.*

We thank the reviewer for bringing this to our attention. Osteoblast specific KO of mTORC1/Raptor leads to several compromised P70S6K activity, bone development and growth (PMID: 28686577). This is similar to the homozygous *aff4*^{-/-} and *rps6kb1/2* crispant zebrafish we report here. We have accordingly modified the text on line 583 in the discussion to reflect the known importance of P70S6K activity on bone development:

“Our data reveal a previously unexplored function of P70S6K in regulating transcriptional elongation, expanding its role beyond its well-characterized actions on protein synthesis, cell survival and skeletal development”⁷⁷.

• *Figure 4J: The knockout of P70S6K has a greater effect on RPS6 than on AFF4. Please discuss whether other kinases may be involved in AFF4 regulation.*

Comparing the effect of P70S6K knockdown on AFF4 vs RPS6 phosphorylation is confounded by a large difference in the abundances of these two phosphopeptides. The RPS6 phosphopeptide was >100-fold more abundant than the AFF4 phosphopeptide (5.4 x 10⁷ vs 3.4 x 10⁵). Since, both precision and accuracy are proportional to abundance, a comparison of responses of different phosphopeptides to the same stimulus is inherently flawed and would be entirely speculative. However, we have commented on the possibility of additional kinases phosphorylating AFF4 in the limitations section on line 621:

“It is also important to note that other kinases other than P70S6K may phosphorylate S831 on AFF4.”

• *RNAP2 traveling ratio (Figures 5H and 5J): There appears to be a discrepancy between the ChIP-seq track area under the curve (AUC) and the bar graph representation of RNAP2 distribution. Clarify how the traveling ratio was calculated and why RNAP2 appears primarily located in the promoter region.*

The travelling ratio is the amount of promotor-paused RNAP2 divided by the amount of RNAP2 in the gene body. Hence, a smaller value is an estimate of more RNAP2 released into productive elongation. Figure 5H/I shows that control cells stimulated with insulin have lower travelling ratio compared to insulin resistant cells suggesting more elongation. This is reflected in Figure 5J, which shows a greater amount of RNAP2 in the gene body in control cells stimulated with insulin compared to insulin resistant cells. Hence, we do not agree that there is a discrepancy.

It is well known that the majority of total RNAP2 remains bound to the promotor region (PMID: 17994021). We apologize for not clearly specifying the calculation of the traveling ratio which is now included on line 1214 in methods:

The traveling ratio was calculated using the method described by¹⁵. Specifically for each gene, we calculated RNAPII traveling index as:

$$\text{RNAPII traveling ratio} = \frac{\text{Read count in TSS region}/L1}{\text{Read count in gene body}/L2}$$

where the transcription start site (TSS) region of a gene is defined as the 50 bp to +300 bp around the TSS and the gene body is defined as +300 bp downstream of the TSS to +3 kb past the transcription end site. Differences in median RNAPII traveling indices were calculated using the Wilcoxon Rank Sum test where the number of asterisks denote the level of statistical significance: ***, p < 0.001; **, p < 0.01; and *, p < 0.05.

• *AFF4 phosphomutants: Multiple serine/threonine residues were mutated to alanine in the mutant AFF4 construct. Are there phosphopeptides in the dataset that support phosphorylation at these sites (e.g., T829, S833, S834, S835)?*

In the mouse bone data, the localization probability of S831 was 82% while S833 was 12% and other sites 1-2%. In HEK293T cells and zebrafish tails, 100% was localized to S831 (S771 equivalent in zebrafish). The PhosphoSitePlus database reports the following number of studies identifying phosphorylation at:

T829: 15

S831: 97

S834: 13

S835: 2

We therefore mutated all the surrounding amino acids adjacent to S831 to prevent potential compensatory phosphorylation or priming phosphorylation of neighboring sites. Multi-site phosphorylation is particularly relevant for P70S6K itself and several known substrates [PMID: 30612880].

• *AFF4 downstream targets (Figure 6): Several downstream targets of AFF4 were identified in HEK293T cells (e.g., Figures 5D–H). Please discuss whether these targets are cell-type specific and what their relevance might be for osteoblast function.*

We have added a brief introduction to the role of EGR1 on osteoblast function given this gene was significantly regulated in both Kusa 4B10 osteoblasts and HEK293T cells. Line 72:

“Furthermore, knockdown of Egr1 in osteoblasts reduces BMP-induced mineralization¹², and EGR1 cooperates with RUNX2 to promote the expression of osteoblastic genes such as Osterix and Osteocalcin¹³.

Reviewer #2 (Remarks to the Author):

This manuscript describes the insulin-stimulated phosphoproteome of bone from old vs young mice. The authors validated several candidates in zebrafish and showed that AFF4 knockout severely affected bone growth in zebrafish. The authors performed a robust validation to show that insulin-stimulated phosphorylation of AFF4-S831 is involved in releasing paused RNAP2 to initiate elongation of select transcripts. This study significantly advances our knowledge of the role of insulin action on AFF4 and release of RNAP2 pausing.

While the validation part of the study to show the role of AFF4 in elongation of select transcripts is very thorough, there are several gaps, such as the criteria for selection of candidates as well as selection of AFF4 for validation. Another gap is the missing link of AFF4 phosphorylation to bone function.

We thank the reviewer for their positive comments and excellent suggestions, and provide point-by-point responses to address gaps and improve the manuscript.

Issues:

The flow of the manuscript is somewhat interrupted by describing phosphoproteomics first, then describing the total proteome, the bone-enriched proteome, the secretome, the enriched transcription factors (but only downregulated proteins), before reverting to the phosphoproteome. No rationale is provided for these diversions, and the only part that may be relevant for the AFF4 part is the enriched transcription factor analysis, but this is not referred to when potentially relevant (after Figure 6). Reordering or even deleting some less relevant parts may help improve the flow. For example, the relevance of the secretome analysis to the rest of the manuscript is not clear unless you can somehow link this to bone function and/or the phosphoproteomic data. E.g. are the proteins mentioned in lines #244-247 bone enriched proteins? And how they relate to the DE phosphoproteins? E.g. Is BMP signalling and WNT signalling also changed in phospho?

The reviewer raises a very good point about the need to improve the flow of the manuscript and/or further highlight the relevance and justify various datasets. We have carefully considered each dataset and modified the manuscript. We thank the reviewer for providing these suggestions to improve the presentation of our study.

The major focus of the manuscript is understanding the signaling mechanisms driving defective insulin-stimulated transcriptional activation in ageing/insulin resistance (IR).

Since our major focus is to understand the signaling mechanisms driving defective insulin-stimulated transcriptional activation in ageing/insulin resistance (IR) we have emphasized this. We have therefore removed all analysis of the secreted proteins from the manuscript including the inferred extracellular proteins regulated in the ageing mouse bones, and the secretome analysis of insulin resistant Kusa 4B10 osteoblasts. We have also edited the manuscript to provide greater rationale for the various transcription factor analysis and highlighted the relevance by comparing the results across the models.

The mouse bone analysis on line 250:

“To understand the potential transcriptional mechanisms controlling the age-associated bone proteome, we next mapped proteins to known upstream transcriptional regulators in the ChEA3db ⁴⁹ and performed GSEA (Figure 2D). Here, we focused on transcriptional regulators that potentially display defective activity and may contribute to decreased protein abundance in aged bone.”

The IR osteoblast analysis starts on line 390:

“IR is associated with defective gene activation in response to acute insulin stimulation ⁶⁶. Under normal healthy states, acute insulin treatment increases mRNA levels of anabolic IEGs such as Jun, Fus and Fos, but the expression of these genes is markedly attenuated under IR states. Given that release of paused RNAP2 and mRNA elongation is a critical checkpoint for gene transcription, we hypothesized that attenuated insulin-dependent phosphorylation of S831 on AFF4 may underlie the transcriptional defects in IR conditions. To investigate this hypothesis, we established an osteoblast IR model by exposing differentiated Kusa 4B10 cells to chronic hyperinsulinemia. First, we characterized the model by quantifying the mineralization capacity, proteome changes and acute insulin signaling responses. Alizarin red staining revealed a significant decrease in the formation of mineralized nodules under IR conditions, a consistent phenotype observed following a reduction in insulin signaling ^{4, 5} (Figure 5A). Proteomic analysis quantified 5,556 protein groups in control or IR osteoblasts and given the emphasis of our investigations is to understand the potential mechanisms of transcriptional defects, we focused on a transcription factor enrichment analysis using the ChEA3db (Figure S2A-C and Table S9). Consistent with our in vivo data, IR osteoblasts displayed a significant reduction in the abundance of known target genes for the transcription factors E2F4 and FOS. These data suggest that IR osteoblasts have a reduced ability to activate IEG.”

While normalising the phosphoproteome to total proteome in principle makes sense, discarding data from 1/3rd of all proteins may omit crucial findings, particularly when focusing on insulin-stimulation. Also the numbers of differentially regulated phosphosites are very confusing in lines #147-150, with 650 or 1004 differentially regulated sites? It is also not clear whether the candidate proteins were identified in the total proteome.

The reviewer is correct that normalizing and assessing changes in phosphopeptide abundance vs total protein abundance may omit important information in cases where total protein quantification is not available. Therefore, we show various comparisons with or without normalization in Table S1 to ensure no data is lost. Here, we show the phosphopeptide fold-changes (insulin vs saline control) of the 10- and 73-week old mice without normalization to total protein abundance. We also show the insulin response of 10- vs 73-week without normalization to total protein levels. We include the 10-week saline vs 73-week saline and the 10-week insulin vs 73-week insulin with and without normalization to total protein levels.

We apologise that the order of describing the number of significantly regulated phosphosites in the various analysis was confusing and we have further clarified the results on line 144:

“To investigate signaling differences between 10- and 73-week-old bone, we first compared the vehicle- and insulin-treated groups. Where possible, this was normalized to the abundance of the total protein levels, since we saw widespread

changes in the proteome between the age groups (see below). Such analysis has previously been performed to compare phosphoproteome versus proteome responses to acute insulin stimulation in the setting of chronic treatments that induce insulin resistance ²⁷. Of the 4,803 phosphoproteins, we obtained proteomic data for 3,195 proteins and normalized the abundance of 15,642 phosphopeptides to the abundance of their protein abundance. In the vehicle-treated groups, there were 65 differentially regulated phosphopeptides between 10-week and 73-week-old mice but when normalized to total protein levels, none of these remained significant ($q < 0.1$; limma moderated t-test with Benjamini Hochberg FDR). This contrasted with the insulin-stimulated tibiae, where 5,471 regulated phosphopeptides differed between 10-week and 73-week-old mice with 650 of these being differentially regulated when normalized to total protein levels. We next compared the magnitudes of the insulin response i.e. difference of vehicle versus insulin stimulation between the age groups. Remarkably, of the 1,418 insulin-regulated phosphosites in young bones, 1,004 showed differences in the magnitude of their response in the 73-week-old mice...

It is surprising that the criteria for candidate prioritisation did not include a change in insulin responsive phosphorylation between young and old, when this was the major aim of the study? The criteria for positive and negative hits in the zebrafish screen are not clear as some of the negatives appeared to have a stronger phenotype than AFF4? In Figure 3B it is difficult to determine which parameters are significant. The selection of AFF4 above any of the other positive hits would benefit from a rationale, particularly since the growth phenotype in Figure 3G was not that marked.

The magnitude of insulin-dependent phosphorylation between 10- and 73-week old mice was indeed a selection criteria and we apologize for not specifying this. We have updated the results on line 280:

“We refined candidate genes that were overlapping in at least two out of the three selection criteria, and further filtered for genes containing differences in their magnitude of insulin-dependent phosphorylation in 10- versus 73-week old mice...”

To clarify the significant data in Figure 3B, we have added asterisks to the bubble plots.

The reviewer is correct that the effect size of the growth of Aff4 F0 crispants, although significant, is smaller than other positive hits from the screen. However, the magnitude in the screen could be due to several reasons such as CRISPR editing efficiency, mosaicism, etc. Hence, we generated F2 germline mutants and observed a much greater response (Figure 3I-K).

We have further expanded on our rationale for selecting AFF4 on line 324:

“These data combined with the previously mentioned clinical features of patients with CHOPS syndrome displaying impairments in skeletal development and obesity caused by missense variants in AFF4 promoted us to further expand our analysis ^{34, 35}”

The calculation of the travelling ratio seems odd as it is counterintuitive that a low travelling ratio reflects less paused RNAP2 (lines 406-410). The data in figure 5F suggests that ~50%

of all RNAP2 transcripts are affected in IR and looks very convincing, but the data presented in 5G-J is much less convincing. Particularly Fig 5G (and also 6C) is difficult to interpret as so many dots (many of which are probably significant) are overlaid and clustered close to zero and maybe alternate visualisation possibilities should be explored.

We have further described the calculation of the travelling ratio to improve the readers interpretation of the data on line 419:

“We calculated the travelling ratio of each gene (amount of promotor-paused RNAP2 / the amount of RNAP2 in the gene body) to estimate the amount of RNAP2 released into productive elongation in the acute insulin-response in control vs IR osteoblasts (Table S10). Here, lower travelling ratios are indicative of more RNAP2 released into productive elongation.”

We have modified Figure 5G to more clearly show that control cells stimulated with insulin generally have lower travelling ratios. We have also modified Figure 6C to a similar format and show that stimulation of cells expressing AFF4-WT generally results in higher fold-changes than cells expressing AFF4-A-mutant. However, this is not true for all genes where several contain a similar level of induction in WT vs A-mutant cells.

Is there any overlap of affected genes between the experiments in Fig 5E-G and Fig 6C?

We have compared the overlap and discussed results on line 459:

“We next compared the RNAP2 ChIP-seq in Kusa 4B10 osteoblasts to the RNA-seq data in HEK293T cells and identified 11 genes as insulin activated in both cell types. Importantly, the regulation of these genes were overall highly similar with genes such as Egr1 and Jun showing attenuated insulin-dependent activation when phosphorylation of S831 on AFF4 was either reduced or ablated whereas genes such as Myc and Atxn713b displayed similar activation regardless of the phosphorylation status.”

Why were Egr1 and Fos selected in Figure 6O?

Egr1 and Fos were selected because they had the highest fold-change following insulin stimulation. Furthermore, EGR1 has previously been shown to play an important role on osteoblast function. We have included an introduction on line 72:

“Furthermore, knockdown of Egr1 in osteoblasts reduces BMP-induced mineralization¹², and EGR1 cooperates with RUNX2 to promote the expression of osteoblastic genes such as Osterix and Osteocalcin¹³.”

Confirmation that the AFF4 – H3K18Cr was prevented with the inhibitor would strengthen the confidence into the inhibitor. Can the role of ELL and ELL2 be excluded as there is a trend for these (significant in Fig 6K but only n=2 data points in Fig 6L)?

The reviewer raises an excellent point and we have performed additional experiments to confirm the YEATS domain inhibitor blocks the interaction of ENL/AF9 with H3K18Cr, and we have included the results in a new Figure 6O and on line 502:

“We next hypothesized that inhibiting the YEATS domain and blocking the interaction of MLLT1/3 of the ENL/AF9 complex with crotonylated histones would attenuate insulin-dependent gene activation. To achieve this, we used the recently developed amido-imidazopyridine inhibitor of the YEATS domain, SR-0815⁶⁹. To test inhibition, nucleosome lysates were generated from cells expressing FLAG-AFF4-WT and treated with or without SR-0815 during anti-FLAG immunoprecipitation (Figure 6O). SR-0815 almost completely blocked the enrichment of K18 crotonylated Histone H3 confirming successful blocking of the YEATS domain.”

We only quantified ELL/ELL2 in two out of the four replicates of the AFF4-WT AP-MS experiments and the reviewer is correct that we cannot exclude a role of ELL/ELL2. However, S831 phosphorylation of AFF4 is directly adjacent to the MLLT1/3 binding site while the ELL/ELL2 binding site is more distal (Figure 4A).

Is there any evidence for a link between histone crotonylation and any AFF4-A affected transcripts?

To investigate potential preliminary links, we re-analysed a recent H3K18Cr ChIP-seq dataset from ESCs which contains ~1800 genes with enriched marks (<https://www.ncbi.nlm.nih.gov/geo/query/acc.cgi?acc=GSE241193>). Out of the 81 genes displaying aberrant insulin-dependent activation in AFF4-A-mutant expressing cells relative to WT cells, 18 displayed enriched H3K18Cr marks. Interestingly, these included some of the most drastically affected genes where mutation of the phosphosite completely blocked insulin-induced activation including *Tbc1d1*, *Wwc1*, and *Ddb2* amongst others. While this provides exciting potential links, we feel including these data are speculative and much further experimental data are required which will significantly extend an already lengthy manuscript.

The model relies on AFF4 phosphorylation to occur prior to recruitment to crotonylated histones via ENL/AF9 but how does this agree with the pausing of the already DNA-bound RNAP2?

This is an interesting question. We hypothesise that: 1- paused RNAP2 is released following insulin activation, 2- transcriptional elongation is then promoted via the activities of P-TEFb and additional kinases such as CDK12/13, 3- P70S6K is subsequently activated to phosphorylate AFF4 and then enhance elongation of specific genes containing crotonylated histone marks. In this model, P70S6K is not directly involved in the early steps of the release of paused RNAP2 but rather acts on a later step where transcriptional complexes are already released or directly adjacent to the promoter. It is important to note that the P70S6K is relatively downstream in the insulin signaling pathway with maximal activation somewhat delayed following insulin stimulation (~10-15 min) compared to other upstream kinases such as Akt (~2 min) (PMID: 23684622). While a high-resolution temporal analysis of AFF4 phosphorylation kinetics and binding to crotonylated histones would provide further insights into this hypothesis, we believe this is beyond the scope of the current manuscript.

How does the AFF4 mediated changes in transcription lead to the bone defects observed in Fig 3G and 3I? E.g. does AFF4 regulate any bone enriched proteins and/or any bone-specific secreted proteins (Fig 2A, 2D)?

This is a critical question and therefore we performed additional experiments by performing a proteomic analysis of zebrafish tails following AFF4 knockdown. This revealed the regulation of several proteins known to be important for skeletal development. For example, Osteomodulin, a proteoglycan important for mineralization was significantly down-regulated following AFF4 knockdown. These data have been included in the results on line 333 and as a new Figure 3 and Supplementary Table 6.

“Proteomic analyses of the caudal fins revealed significant regulation of 237 proteins in the aff4 knockout fish when compared to the scr control ($q < 0.05$; Benjamini-Hochberg correction)(Figure 3L & Table S6). Pathway enrichment analysis using FishEnrichr revealed an up-regulation of proteins involved in ubiquitin-dependent proteolysis and associated peptide catabolic processes (Figure 3M). These include the proteasome activator complex subunits 1/2 (PSME1/2) and the interferon-induced GTP-binding protein class (MXA, MXC, MXE). Knockdown of aff4 was associated with the down-regulation of proteins involved in VEGF signaling, ECM organization and the development of the skeleton and fin morphology such as osteomodulin (OMD) and myocilin (MYOC), both positive regulators of osteoblast differentiation, along with type II collagens (COL2A1A and COL2A1B) that are required for skeletal growth. Additionally, proteins involved in vascular endothelial growth factor signalling important for angiogenesis (neuropilins, NRP1A, NRP1B, NRP2A ; BMP-binding endothelial regulator protein precursor, BMPER), and systemic muscle development (myosin heavy chain 7, MYH7; cardiac troponin T type 2B, TNN2B) were observed to be significantly downregulated, suggesting that aff4 knockdown fish have a compromised musculoskeletal development when compared to the scr.”

Minor issues:

The rib figure was a bit confusing as 18+ maybe is meant to be <18?

We thank the reviewer for bringing this typo to our attention. Indeed, the bar graph in Figure 3G should be <18 and not 18+. We have corrected this in the revised Figure 3.

Please explain why the scr fish differ so vastly in length (3G vs 4G).

Zebrafish development occurs externally, and its growth rate is influenced by a range of environmental factors such as water temperature and quality, type and frequency of feeding, and housing conditions such as tank size and density (PMID: 19891001, PMID: 24979389). Furthermore, the genetics of the parent zebrafish (being an outbred model) also affect growth rate and batches of fish as different breeding parents can exhibit a faster or slower growth rate; this likely explains why the scr fish are of different sizes. To control for this in our functional genomic screening assay, the candidate gene and negative scr CRISPR/Cas9 injections were always performed in parallel on the same embryo clutch generated from the same parent pair, and the

candidate gene was directly compared to its sibling negative scr control for statistical analyses.

We have modified lines 1322 under methods to provide a brief comment on zebrafish growth rate and to remove any ambiguity as follows:

“As our zebrafish are outbred, their growth depends on the genetics of the parent fish combined with environmental factors. For direct comparison, the candidate gene and negative scr injections were performed in parallel from the same embryo clutch, generated from the same breeding pair.”

Reviewer #1 (Remarks to the Author):

The revised manuscript has been edited for improved clarity. As requested, new experimental data have been included, and the authors have addressed the issues raised by this reviewer. The authors have strengthened their case, and the manuscript has improved considerably. I have no further comments.

Reviewer #2 (Remarks to the Author):

The authors have addressed all issues raised by this reviewer. They have improved the manuscript including the flow, and clarified several sections including the selection criteria. The authors could consider adding a sentence or two at line 563 in the discussion on the crotonylation status of the proteins that were downregulated in the AFF4 knockdown zebrafish, if such information is available, as an additional link of AFF4 phosphorylation to bone function. But this is just a suggestion and the manuscript is good as it is.

Given the length of the manuscript, speculating on the crotonylation status of individual genes is beyond the scope of this manuscript.

Minor issues:

Track changes on line 199.

Track change on line 199 has been fixed.

Typo in line 604 (dependent)

Typo in line 604 has been corrected to “dependent”.

Reviewer #3 (Remarks to the Author):

The manuscript by Dutt and Colleagues describes the role of AFF4, a component of the super elongation complex (SEC), in mediating insulin-regulated gene expression. By comparing the bone phosphoproteomes of young and aged insulin-resistant (IR) mice, the authors identified a list of candidate proteins whose phosphorylation levels are significantly different between the two groups. To assess the role of these candidates in bone biology, they disrupted the orthologous genes in zebrafish and identified several genes, including *aff4*, whose knockdown leads to bone defects. The author went on to demonstrate that AFF4 S831 phosphorylation is blunted in an IR cell model. They determine that P70S6K mediates the insulin-dependent phosphorylation of AFF4 S831 in HEK293T cells, and this phosphorylation is necessary for promoting the expression of insulin-regulated genes HEK293T and Kusa 4B10 cells. They further show that AFF4-pS831 recruits ENL/AF9 complex to the SEC to enhance transcription. Therefore, the study identified a new mechanism of insulin-regulated gene expression via modulating transcriptional elongation and demonstrated its deficiency in IR. The methodologies are sound, the results are of good quality, the conclusions are well supported, and the manuscript is clearly written. Nevertheless, there are a few issues.

Major issues:

- In the zebrafish studies, the mutagenesis rate of each sgRNA is not characterized. Although efficient mutagenesis of some sgRNAs is evident in Fig S4, the efficiency of most sgRNA cannot be discerned from the images. A TIDE analysis may be more informative. Such results would provide the readers with more confidence that the negative hits are true negatives. In the same vein, it would be great if the authors can provide the exact mutation in the germline *aff4* line they further characterized.

We thank the reviewer for raising this point. We qualitatively assessed successful editing of all target genes via PCR. However, we did not assess the mutagenesis rate of each sgRNA via Sanger sequencing and therefore we are not able to perform a TIDE analysis as suggested. Given our focus was AFF4, we only validated mutagenesis of the *aff4* F2 crispants by Sanger sequencing and have now included the exact germline mutations in Supplementary Figure 5 as requested.

Minor issues:

- It is unclear why the *aff4* is prioritized in the zebrafish study. The KD phenotype of the *aff4* crispants is the weakest among the positive hits. In fact, it seems to be weaker than some of the negative hits.

AFF4 was prioritised because:

- 1- S831 was a top predicted Akt/P70S6K substrate**
- 2- S831 had attenuated insulin-dependent phosphorylation in old, insulin resistant (IR) bones of mice**
- 3- S831 had attenuated insulin-dependent phosphorylation in IR osteoblasts**
- 4- Loss-of-function mutations in AFF4 are associated with CHOPS syndrome which display clinical features of skeletal dysplasia and obesity.**

- 5- GWAS have identified variants in *AFF4* that are associated with reduced bone mineral density.
- 6- *Aff4* F0 crispants displayed significant reductions in body length and rib count.
- 7- *Aff4* germline mutants displayed significant reductions in body length, mineralisation and skeleton length.

• *In the experiments presented in Figure 6, AFF4 mutant cells would be a better model for the overexpression of the WT and mutant AFF4.*

We attempted to mutate endogenous S831 via CRISPR HDR but we were unable to obtain a successful stable cell line. Hence, we proceeded with an over-expression model noting that the exogenous transgene likely competes with endogenous AFF4 during the assembly of the Super Elongation Complex.